# ACTION-AWARE DYNAMIC PRUNING FOR EFFICIENT VISION-LANGUAGE-ACTION MANIPULATION

**Xiaohuan Pei**[1]*, **Yuxing Chen**[1]*, **Yuheng Shi**[1], **Siyu Xu**[1], **Yunke Wang**[1], **Chang Xu**[1]†
[1]Department of Computer Science, The University of Sydney, Australia
`xiaohuan.pei@sydney.edu.au   yche0009@uni.sydney.edu.au`
`{yuheng.shi, s.xu, yunke.wang, c.xu}@sydney.edu.au`

## ABSTRACT

Robotic manipulation with Vision-Language-Action models requires efficient inference over long-horizon multi-modal context, where attention to dense visual tokens dominates computational cost. Existing methods optimize inference speed by reducing visual redundancy within VLA models, but they overlook the varying redundancy across robotic manipulation stages. We observe that the visual token redundancy is higher in coarse manipulation phase than in fine-grained operations, and is strongly correlated with the action dynamic. Motivated by this observation, we propose **A**ction-aware **D**ynamic **P**runing (**ADP**), a multi-modal pruning framework that integrates text-driven token selection with action-aware trajectory gating. Our method introduces a gating mechanism that conditions the pruning signal on recent action trajectories, using past motion windows to adaptively adjust token retention ratios in accordance with dynamics, thereby balancing computational efficiency and perceptual precision across different manipulation stages. Extensive experiments on the LIBERO suites and diverse real-world scenarios demonstrate that our method significantly reduces FLOPs and action inference latency (*e.g.* 1.35× speedup on OpenVLA-OFT) while maintaining competitive success rates compared to baselines, thereby providing a simple plug-in path to efficient robot policies that advances the efficiency and performance frontier of robotic manipulation. Project is available at ADP.

## 1 INTRODUCTION

Large vision language models Liu et al. (2023c;b; 2024a); Team et al. (2023); Awadalla et al. (2023) have recently been extended into **V**ision-**L**anguage-**A**ction (VLA) models Kim et al. (2024; 2025); Black et al. (2024); Li et al. (2024); Brohan et al. (2024); Wen et al. (2025b;a); Bjorck et al. (2025) that map both the visual observation and language instruction to executable robot actions. In the mainstream pipeline, a vision encoder produces dense visual tokens from one or more camera views, a projector aligns them to the language space, and an LLM fuses all modalities to predict actions. However, this multi-modal design introduces long input sequences with numerous visual tokens that are only weakly relevant to the current manipulation operation, which inflates compute, memory footprint, and latency, and it can dilute attention over truly task-relevant cues.

Existing work pursues efficiency via architectural lightening and modality-aware compression, such as RoboMamba Liu et al. (2024b) that focuses on lightweight designs, DeeR-VLA Yue et al. (2024) that aims at structured pruning/reparameterization, Mole-VLA Zhang et al. (2025) that targets conditional layer activation, VLA-Cache Xu et al. (2025b) that focuses on cache reuse, and EfficientVLA Yang et al. (2025) that aims to prune visual tokens via attention. However, a key but underexplored property of robotic manipulation is that **visual redundancy in VLAs is action-aware across different manipulation stages**. As Figure 1 shows, during coarse-grained operations (*e.g.*, relocating), global movement dominates and redundant tokens can be pruned; during fine-grained phases (*e.g.*, grasping), local geometry and detailed cues dominate and preserving full vision is preferred. Moreover, the relevance of visual patches is not only text conditioned (semantics of the instruction) but also *action*

---

*Equal contributions
†Corresponding author.

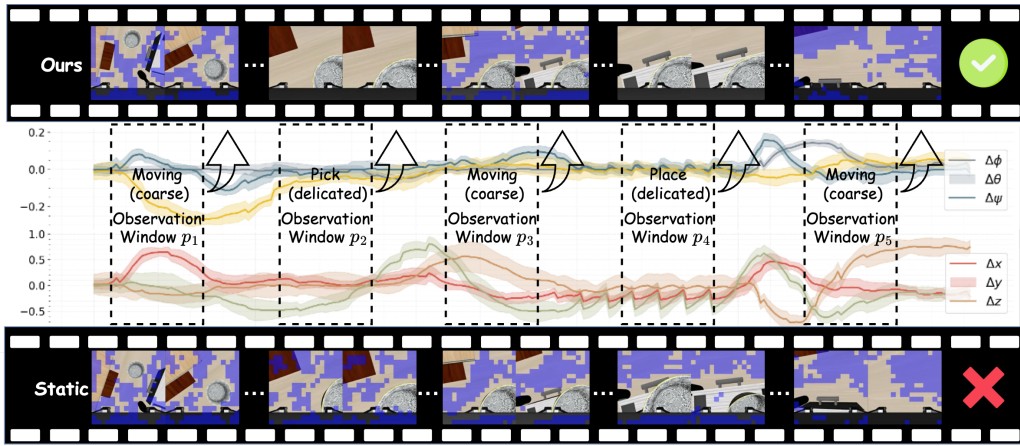

Figure 1: **Action-aware dynamic pruning vs. static pruning.** We visualize five *past* observation windows as cases from a manipulation episode that condition the *current anticipatory* window. $p_1$, $p_3$ and $p_5$ reflect coarse phases, prompting the gate to *enable* pruning to suppress redundant tokens, whereas $p_2$ and $p_4$ are delicate phases requiring detail vision context, so pruning is *disabled* and full vision is used. The curves depict robot's motion that drives the gating rule.

*conditioned* (instantaneous end-effector motion and gripper state). Treating all steps uniformly, or ranking tokens solely by mixed attention scores, therefore yields suboptimal pruning schedules that either prune too little (limited savings) or prune too much (accuracy loss), especially in multi-view settings (scene and wrist/gripper cameras) where importance is unevenly distributed across time.

To address this challenge, we introduce Action-aware Dynamic Pruning (ADP), a plug-and-play strategy that reduces computation while preserving manipulation fidelity. ADP is built on two complementary ideas: (1) *Text-driven Pruning* evaluates the relevance of visual patch using cross-modal similarities, selecting only the most relevant tokens before entering deep fusion in the subsequent layers. (2) *Action-aware Dynamics* modulate whether pruning is activated at a given step using a lightweight decision signal derived from the end-effector trajectory within each action window. Specifically, when the recent motion magnitude is relatively low compared to past motion statistics (delicate phases), pruning is disabled to preserve the full visual field for precise control. Conversely, when the motion magnitude is relatively high compared to past motion statistics (coarse phases), pruning is engaged to suppress redundancy and save FLOPs. We implement a gated mechanism that treats recent action statistics as a pruning signal over sliding trajectory windows, adaptively adjusting retention ratios according to motion dynamics and balancing efficiency with precision across manipulation stages. Our contributions can be summarized as: (1) We show that the importance of visual tokens in VLA models varies within different stages of robotic manipulation. This insight motivates our dynamic pruning method tailored to manipulation phases compared to static pruning approaches. (2) We propose *text-driven action-aware pruning* that combines task instruction relevance with a gating rule based on end-effector motion, enabling adaptive switching between pruned and full-vision states. (3) We present a principled complexity analysis and extensive experiments in both simulation and real-world settings, demonstrating that our method reduces FLOPs and latency while maintaining fine visual details required for successful manipulation.

## 2 PRELIMINARY

The mainstream vision-language-action paradigm extends large vision-language models to generate executable robot actions from multi-modal inputs—visual observations (scene and wrist/gripper views) and task instructions. A pre-trained vision encoder produces visual tokens, a projector aligns them to the LLM token space, and the LLM fuses modalities and autoregressively emits action tokens, which are de-tokenized into a continuous 7-dimensional robot action.

Formally, given a sample of a scene image $I^s \in \mathbb{R}^{H \times W \times 3}$, a gripper view $I^g \in \mathbb{R}^{H \times W \times 3}$, and a task instruction $I^t \in \mathbb{R}^{N \times L}$, the vision encoder $f_{\text{enc}}^v$ (DINOv2 Oquab et al. (2023) and SigLIP Zhai et al. (2023)) and text tokenizer $f_{\text{enc}}^t$ project the multi-modal data into the same latent dimension space:

$$\mathbf{X}^{\text{vis}} = f_{\text{enc}}^v(I^s, I^g), \quad \mathbf{X}^{\text{txt}} = f_{\text{enc}}^t(I^t), \tag{1}$$

where $\mathbf{X}^{\text{vis}} \in \mathbb{R}^{L_{\text{vis}} \times D}$ and $\mathbf{X}^{\text{txt}} \in \mathbb{R}^{L_{\text{txt}} \times D}$ represent latent embeddings for vision and text.

**OpenVLA.** In the latent space, the embedding representations are concatenated into a multi-modal sequence as:

$$\mathbf{X}^m = \mathbf{X}^{[BOS]} \oplus \mathbf{X}^{\text{vis}} \oplus \mathbf{X}^{\text{prop}} \oplus \mathbf{X}^{\text{txt}} \tag{2}$$

where $\oplus$ denotes concatenating embeddings of [BOS], vision, text, and proprioceptive (optionally) along the sequence length dimension, yielding multi-modal inputs $\mathbf{X}^m \in \mathbb{R}^{1+L_{\text{vis}}+L_{\text{txt}}+1}$. The multi-modal sequence $\mathbf{X}^m$ is then fed into a Large Language Model (LLM) $f_{\text{LLM}}$ (Llama2 Touvron et al. (2023)), which performs contextual reasoning and autoregressively generates an action token sequence:

$$p(\hat{\mathbf{a}} \mid \mathbf{X}^m) = \prod_{j=1}^{7} f_{\text{LLM}}(\hat{a}_j \mid \mathbf{X}^m, \hat{a}_{<j}), \tag{3}$$

where $\hat{\mathbf{a}} = (\hat{a}_1, \ldots, \hat{a}_7) \in \mathbb{T}^7$ and $\mathbb{T} = \{1, \ldots, K\}$ denotes the reserved action token space.

**OpenVLA-OFT.** For optimize finetuning version, the paradigm shifts to parallel decoding without traditional autoregressive decoding stage. In the latent space, the method introduces $L_{\text{act}}$ placeholder $\mathbf{X}^{\text{place}}$ as inputs for actions positions:

$$\mathbf{X}^m = \mathbf{X}^{[BOS]} \oplus \mathbf{X}^{\text{vis}} \oplus \mathbf{X}^{\text{prop}} \oplus \mathbf{X}^{\text{txt}} \oplus \mathbf{X}^{\text{place}}, \tag{4}$$

where $\mathbf{X}^m \in \mathbb{R}^{1+L_{\text{vis}}+L_{\text{txt}}+1+L_{\text{act}}}$ and the first placeholder as current action placeholder and another $L_{\text{act}} - 1$ actions as future actions. During generation, the LLM directly predicts action tokens at action chunk in parallel:

$$\hat{\mathbf{A}} = f_{\text{LLM}}(\mathbf{X}^m)\big|_{\text{place}} \in \mathbb{T}^{L_{\text{act}} \times 7}, \tag{5}$$

where $\hat{\mathbf{A}} = [\hat{\mathbf{a}}_1; \ldots; \hat{\mathbf{a}}_{L_{\text{act}}}]$, and each $\hat{\mathbf{a}}_i = (\hat{a}_{i,1}, \ldots, \hat{a}_{i,7}) \in \mathbb{T}^7$ represents the tokenized 7-DoF action predicted at the $i$-th placeholder position. To expand action representation ability, the oft version also introduces the continuous control with token-level modeling, which each dimension of the continuous action is uniformly discretized into 256 bins. During generation, the 7 degrees of freedom action follows the standard gripper parameterization:

$$\hat{\mathbf{a}}^c = [\,\Delta x,\ \Delta y,\ \Delta z,\ \Delta\phi,\ \Delta\theta,\ \Delta\psi,\ g\,]^\top, \tag{6}$$

where $(\Delta x, \Delta y, \Delta z)$ denote Cartesian displacements, $(\Delta\phi, \Delta\theta, \Delta\psi)$ denote Euler-angle rotations (roll, pitch, yaw), and $g$ denotes the gripper.

## 3 RELATED WORK

**Vision-Language-Actions.** Vision-Language-Action (VLA) models extend large vision-language models with an action generator that links multimodal perception to low-level robot control. They take images and text instructions, encode them into a shared latent space, then decode it into executable actions. Early approaches Chi et al. (2023); Kim et al. (2024) map multimodal embeddings to discrete tokens, while more recent works Wen et al. (2025b); Kim et al. (2025) emphasize continuous parallel decoding actions for improving performance. To this end, lightweight MLPs, diffusion-based decoders, and parallel decoding strategies have been introduced to transform hidden states into temporally consistent trajectories. Representative architectures include CogACT Li et al. (2024), OpenVLA Kim et al. (2024), OpenVLA-OFT Kim et al. (2025), and $\pi$ series Black et al. (2024), which adopt diffusion modules for iterative refinement of continuous actions, as well as optimized fine-tuning frameworks that leverage placeholder tokens for parallel action prediction.

**Efficient Robotic Manipulations.** The high computational complexity of VLA models poses significant efficiency challenges for real-time robotic control. To address this, recent works have proposed efficiency-oriented strategies that can be broadly divided into training-aware and training-free approaches. Training-aware methods, such as RoboMamba Liu et al. (2024b) and DeeR-VLA Yue et al. (2024), redesign architectures or apply compression and pruning during training, achieving notable speedups while preserving accuracy. For instance, DeeR-VLA Yue et al. (2024) introduces dynamic reparameterization and structured pruning to reduce FLOPs, and Mole-VLA Zhang et al. (2025) selectively activates subsets of model layers conditioned on task demands, yielding scalable deployment. On the other hand, training-free methods aim to accelerate inference without retraining.

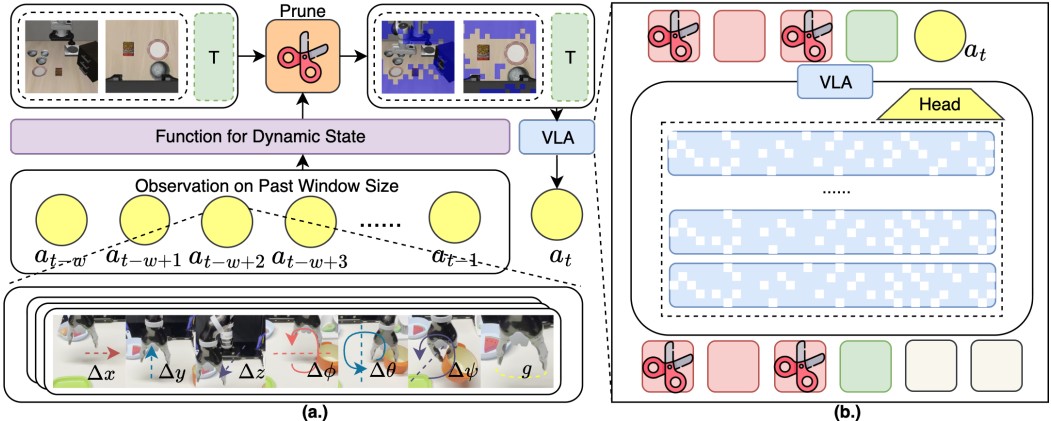

Figure 2: Overview of our proposed **A**ction-aware **D**ynamic **P**runing (**ADP**) for Vision-Language-Action models. **(a.)** Action-aware gating: the pruning function adaptively determines whether to prune based on recent end-effector trajectories $(\Delta x, \Delta y, \Delta z, \Delta \phi, \Delta \theta, \Delta \psi, g)$, enabling dynamic pruning. **(b.)** Anticipatory pruning: task-relevant visual tokens are selected via attention-based relevance, while redundant patches are discarded before entering the VLA backbone.

VLA-Cache Xu et al. (2025b) reuses keys and values of uninformative tokens between consecutive steps to save computation, while EfficientVLA Yang et al. (2025) retain task-relevant patches identified via attention maps. However, these methods typically rely on single-layer heuristics or static rules that may overlook stage-dependent redundancy in manipulation tasks.

## 4 METHODOLOGY

In this section, we introduce our proposed method Action-aware Dynamic Pruning (ADP) in VLAs for robotic manipulations. We first introduce text-driven pruning that identifies text-relevant tokens in Section 4.1 and then introduce action-aware dynamics that dynamically modulates the pruning strategy according to the past observed actions in Section 4.2.

### 4.1 TEXT-DRIVEN ANTICIPATORY PRUNING

Before entering the LLM, the multimodal sequence $\mathbf{X}$ still contains a large number of redundant visual tokens, which increase computation and dilute attention focus. To mitigate this redundancy, we compute the relevance of visual tokens with respect to task instructions at each layer, which is shown in Figure 3. Let the hidden state at layer $l$ be $\mathbf{H}^{(l)} \in \mathbb{R}^{S \times D}$ ($\mathbf{H}^{(0)} = \mathbf{X}^m$ in Eq. 2). We partition $\mathbf{H}^{(l)}$ into the vision and text subsets $\mathbf{H}_{\text{vis}}^{(l)} \in \mathbb{R}^{L^{\text{vis}} \times D}$ and $\mathbf{H}_{\text{txt}}^{(l)} \in \mathbb{R}^{L^{\text{txt}} \times D}$. Applying the projection matrices, we obtain query and key representations:

$$\mathbf{Q}^{(l)} = \mathbf{H}_{\text{txt}}^{(l)} W_Q^{(l)}, \quad \mathbf{K}^{(l)} = \mathbf{H}_{\text{vis}}^{(l)} W_K^{(l)}, \tag{7}$$

reshaped into multi-head form $\mathbf{Q}^{(l)} \in \mathbb{R}^{N^h \times L^{\text{txt}} \times d}$ and $\mathbf{K}^{(l)} \in \mathbb{R}^{N^h \times L^{\text{vis}} \times d}$. The scaled dot-product similarity is then computed as

$$\mathbf{A}^{(l)} = \frac{\mathbf{Q}^{(l)}(\mathbf{K}^{(l)})^\top}{\sqrt{d}} \in \mathbb{R}^{N^h \times L^{\text{txt}} \times L^{\text{vis}}}, \tag{8}$$

where each entry measures the degree to which a text token attends to a visual patch. To derive a global importance score per visual token, we average across heads and text queries:

$$\Phi^{(l)}(v) = \frac{1}{N^h \cdot L_{\text{txt}}} \sum_{h=1}^{N^h} \sum_{t=1}^{L_{\text{txt}}} \mathbf{A}_{h,t,v}^{(l)}, \tag{9}$$

$$\mathbf{X}_{\text{keep}} = \text{Top-K}\left(\Phi^{(l)}, k\right), \quad k = \lfloor \rho \cdot L^{\text{vis}} \rfloor, \tag{10}$$

where $v \in \{1, \ldots, L_{\text{vis}}\}$ indexes visual tokens, $N^h$ is the number of attention heads, $L^{\text{txt}}$ is the text sequence length, and $L^{\text{vis}}$ is the total number of visual tokens. In multi-view scenarios with $C$ input images (*e.g.*, scene and wrist views), each contributing $L_c^{\text{vis}}$ patches such that $\sum_{c=1}^{C} L_c^{\text{vis}} = L^{\text{vis}}$, the retention quota is distributed across views by a weighting vector $\alpha \in \mathbb{R}^C$ with $\sum_{c=1}^{C} \alpha_c = 1$. The remaining visual tokens kept for view $c$ are

$$\mathbf{X}_{(c)}^{\text{vis}} = \text{Top-K}\left(\Phi_{(c)}^{(l)}, k_c\right), \quad k_c = \lfloor \alpha_c \cdot k \rfloor, \tag{11}$$

where $\Phi_{(c)}^{(l)}$ denotes the importance scores restricted to view $c$. For each view $c$, the remaining visual tokens can be represented as:

$$\mathbf{X}_{\text{keep}}^{\text{vis}} = \bigcup_{c=1}^{C} \{\, \mathbf{X}_{(c)}^{\text{vis}}[v] \;|\; v \in \mathbf{X}_{(c)}^{\text{vis}}, \; \mathbf{X}_{(c)}^{\text{vis}}[v] \in \mathbb{R}^D \,\}. \tag{12}$$

The reduced visual sequence $\mathbf{X}_{\text{keep}}^{\text{vis}}$ is then concatenated back with other modalities to form the pruned multimodal sequence:

$$\tilde{\mathbf{X}}^m = \mathbf{X}^{[\text{BOS}]} \oplus \mathbf{X}_{\text{keep}}^{\text{vis}} \oplus \mathbf{X}^{\text{prop}} \oplus \mathbf{X}^{\text{txt}} \oplus \mathbf{X}^{\text{act}} \oplus \mathbf{X}^{[\text{EOS}]}, \tag{13}$$

where $\tilde{\mathbf{X}}^m$ denotes the dynamically reduced input propagated to the LLM module.

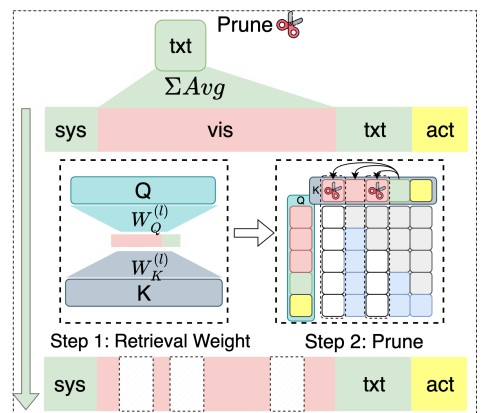

Figure 3: Text-driven Anticipatory Pruning. **Step 1**: Retrieval pretrained weights from Layer $l$ to compute relevance scores. **Step 2**: Treat text as a guide to prune vision tokens based on the ranking .

## 4.2 ACTION-AWARE DYNAMIC STRATEGY

While static pruning can effectively identify task-relevant visual tokens, not every stage of a manipulation task is suitable for relying solely on the pruned set. In particular, missing fine-grained visual details may cause failures in operations such as moving, swiping, or aligning objects. The accumulation of such local errors can easily propagate and ultimately cause the entire task to fail. To address this limitation, we introduce a dynamic visual strategy that adapts the pruning decision to the robot's motion state across different task phases, guided by the end-effector (EEF) trajectory within each action chunk, which focuses on capturing both translational displacement and rotational motion.

**Windowed trajectory and actions.** Assume $i$ indexes windows and $u$ indexes steps within a window; $b_i$ and $e_i$ denote the start and end timesteps, and $\omega$ matches the OFT action placeholder length. We treat each decoded action chunk as a temporal window $[b_i, e_i]$ of length $\omega = e_i - b_i + 1$ with actions $\mathbf{A}_i^c = [\mathbf{a}_{i,1}^c; \ldots; \mathbf{a}_{i,\omega}^c] \in \mathbb{R}^{\omega \times 7}$, where $\mathbf{a}_{i,u}^c = [\Delta x_{i,u}, \Delta y_{i,u}, \Delta z_{i,u}, \Delta\phi_{i,u}, \Delta\theta_{i,u}, \Delta\psi_{i,u}, g_{i,u}]^\top$ collects per-step translational and rotational increments and the gripper command.

**Definition 4.1** (Windowed FK for EEF Position). *Let $T_t \in SE(3)$ be the EEF pose at time $t$, and $\pi : SE(3) \rightarrow \mathbb{R}^3$ extract its translation. With body-frame (right-multiply) composition,*

$$\mathbf{p}_{b_i+u} \triangleq \pi\left(T_{b_i} \prod_{k=1}^{u} \begin{bmatrix} R_{i,k} & \mathbf{v}_{i,k} \\ \mathbf{0}^\top & 1 \end{bmatrix}\right), \qquad T_{b_i+u} \triangleq T_{b_i+u-1} \begin{bmatrix} R_{i,u} & \mathbf{v}_{i,u} \\ \mathbf{0}^\top & 1 \end{bmatrix}, \tag{14}$$

*where $\mathbf{v}_{i,u} = [\Delta x_{i,u}, \Delta y_{i,u}, \Delta z_{i,u}]^\top$ and $R_{i,u} = R_x(\Delta\phi_{i,u})\, R_y(\Delta\theta_{i,u})\, R_z(\Delta\psi_{i,u})$.*

The rotation order $R_x R_y R_z$ follows our implementation; if a world-frame (left-multiply) update or a different Euler order is used, the composition should be adjusted accordingly. Definition 4.1 therefore makes $\mathbf{p}_t = \pi(T_t)$ an explicit function of the full 7-DoF action sequence within each window.

**Windowed trajectory distance.** Given $\mathbf{p}_t$ from Definition 4.1, we quantify motion per window either by the Euclidean displacement:

$$\delta_i = \sum_{t=b_i}^{e_i-1} \|\mathbf{p}_{t+1} - \mathbf{p}_t\|_2. \tag{15}$$

The windowed trajectory distance aims to yield a scalar $\delta_i$ that captures overall motion magnitude and subsequently drives our pruning decision rule.

**Dynamic decision function.** Given a windowed trajectory distance, we define a binary state variable $s_i \in \{0, 1\}$, where $s_i = 0$ corresponds to the *full-vision* state (w/o pruning) and $s_i = 1$ to the *pruned* state (cross-attention pruning). Therefore, the next state can be determined by

$$s_{i+1} = f(\delta_i) = \begin{cases} 1, & \delta_i \geq \bar{\delta}_i, \\ 0, & \delta_i < \bar{\delta}_i; \end{cases} \qquad \bar{\delta}_i = \frac{1}{i} \sum_{j=1}^{i} \delta_j. \tag{16}$$

The dynamic switch to $s_{i+1}$ adapts to the global motion scale of the task, enabling pruning during periods of high activity while retaining full vision during fine-grained phases. An alternative is the *adjacent-extrema function*, which sets thresholds based on the extrema of the most recent $\tau$ windows:

$$U^{(i)} = \max\{\delta_{i-\tau+1}, \ldots, \delta_i\},$$
$$V^{(i)} = \min\{\delta_{i-\tau+1}, \ldots, \delta_i\} \tag{17}$$

with the update rule

$$s_{i+1} = \begin{cases} 1, & \delta_i \geq U^{(i)}, \\ 0, & \delta_i \leq V^{(i)}, \\ s_i, & V^{(i)} < \delta_i < U^{(i)}. \end{cases} \tag{18}$$

This design aims to respond quickly to local motion variations, capturing abrupt efficient shifts between coarse and delicate manipulations. When the robot exhibits large-amplitude motions, pruning is activated to suppress redundant visual input and reduce computation. When motion amplitude decreases, signaling the onset of fine manipulation, pruning is disabled to preserve the complete visual context necessary for accurate control.

## 4.3 THEORETICAL ANALYSIS OF COMPUTATIONAL COMPLEXITY

We consider the computational cost of the Transformer stack in $f_{\text{LLM}}$ with respect to the sequence length before and after pruning. Let $S$ denote the token length of the full multi-modal sequence in Eq. 2, $D$ the hidden dimension, and $M$ the intermediate dimension of the feed-forward network (SwiGLU). The total number of vision tokens before pruning is $L_{\text{vis}}$, while $k = \lfloor \rho \cdot L_{\text{vis}} \rfloor$ tokens are kept after the text-driven selection in Eq. 10. We denote by $T$ the number of forwards in a complete task execution (Eq. 5), and by $\gamma \in [0, 1]$ the proportion of forwards executed in the pruned state (Section 4.2). Let $H$ be the number of Transformer layers in $f_{\text{LLM}}$.

For one Transformer layer, the FLOPs are approximated by

$$F(S; D, M) \approx 2S^2 D + 8SD^2 + 6SDM, \tag{19}$$

corresponding to attention, projections, and MLP. A baseline forward of the $H$-layer LLM therefore costs

$$F_{\text{base}} = H \cdot F(S; D, M). \tag{20}$$

In ADP, pruning happens *before* the LLM at the embedding stage: visual tokens are ranked and reduced using Eqs. 9-10, yielding the shorter sequence

$$S' = 1 + k + L_{\text{prop}} + L_{\text{txt}} + L_{\text{act}} + 1. \tag{21}$$

The scoring overhead uses lightweight projections and a similarity matrix between text and vision embeddings,

$$F_{\text{score}} = 2L_{\text{txt}}D^2 + 2L_{\text{vis}}D^2 + 2N^h L_{\text{txt}} L_{\text{vis}} d. \tag{22}$$

Assume pruning and scoring are performed once per forward *before* any Transformer layer, so all $H$ layers operate on $S'$, the per-layer cost follows $F(S; D, M) \approx 2S^2 D + 8SD^2 + 6SDM$, and $D = N^h d$,

$$\Delta F_{\text{ADP}} = F_{\text{base}} - F_{\text{ADP}}, \quad F_{\text{ADP}} = F_{\text{score}} + H \cdot F(S'; D, M), \tag{23}$$

where $k = \lfloor \rho L_{\text{vis}} \rfloor$, $S'$ is as above with $L_{\text{prop}}, L_{\text{txt}}, L_{\text{act}}$ the proprioception/text/action lengths, $N^h$ is the number of attention heads of size $d$, and $F_{\text{base}} = H F(S; D, M)$. Over an episode of $T$ forwards, the expected complexity under the dynamic strategy is

$$\mathbb{E}[\mathcal{F}_{\text{episode}}] = T\big(\gamma F_{\text{ADP}} + (1 - \gamma) F_{\text{base}}\big), \tag{24}$$

Table 1: **Results on the LIBERO Benchmark. TR**: Training-Free; **AR**: Auto-Regressive; **PD**: Parallel Decoding. **Ratio**: The retain tokens / Full tokens.

| Method | TR | Decoding | CKPT | Spatial | Object | Goal | Long | Average | FLOPs↓ | Speedup↑ |
|---|---|---|---|---|---|---|---|---|---|---|
| OpenVLA Kim et al. (2024) | - | AR | OpenVLA (7B) | 84.7% | 88.4% | 79.2% | 53.7% | 76.5% | - | - |
| SparseVLM Zhang et al. (2024) | ✓ | AR | OpenVLA (7B) | 79.8% | 67.0% | 72.6% | 39.4% | 64.7% | - | - |
| FastV Chen et al. (2024) | ✓ | AR | OpenVLA (7B) | 83.4% | 84.0% | 74.2% | 51.6% | 73.3% | | - |
| VLA-Cache Xu et al. (2025b) | ✓ | AR | OpenVLA (7B) | 83.8% | 85.8% | 76.4% | 52.8% | 74.7% | - | - |
| FlashVLA Tan et al. (2025) | ✓ | AR | OpenVLA (7B) | 84.2% | 86.4% | 75.4% | 51.4% | 74.4% | - | - |
| SP-VLA Li et al. (2025) | ✓ | AR | OpenVLA (7B) | 75.4% | 85.6% | 84.4% | 54.2% | 74.9% | - | - |
| WorldVLA Cen et al. (2025) | - | AR | Chameleon (7B) | 85.6% | 89.0% | 82.6% | 59.0% | 79.1% | - | - |
| WorldVLA* Cen et al. (2025) | - | AR | Chameleon (7B) | 87.6% | 96.2% | 83.4% | 60.0% | 81.8% | - | - |
| NORA Hung et al. (2025) | - | AR | Qwen-VL (3B) | 85.6% | 87.8% | 77.0% | 45.0% | 73.9% | - | - |
| SmolVLA Shukor et al. (2025) | - | AR | SmolVLM (2.25B) | 93.0% | 94.0% | 91.0% | 77.0% | 88.8% | - | - |
| CogACT Li et al. (2024) | - | FM | CogVLM (7B) | 97.2% | 98.0% | 90.2% | 88.8% | 93.6% | - | - |
| NORA-Long Hung et al. (2025) | - | PD | Qwen-VL (3B) | 92.2% | 95.4% | 89.4% | 74.6% | 87.9% | - | - |
| OpenVLA-OFT Kim et al. (2025) | - | PD | OFT (7B) | 98.6% | 98.2% | 94.8% | 94.8% | 97.1% | 7.91 | 1.00× |
| FastV(+OFT) Chen et al. (2024) | ✓ | PD | OFT (7B) | 96.8% | 81.0% | 96.4% | 73.0% | 86.8% | 6.37 | 1.24× |
| **VLA-ADP (Ratio=30%)** | ✓ | PD | OFT (7B) | 97.6% | **98.4%** | **97.4%** | 84.2% | 94.4% | 5.85 | 1.35× |
| **VLA-ADP (Ratio=40%)** | ✓ | PD | OFT (7B) | 98.2% | 97.2% | 96.6% | 87.2% | 94.8% | 6.14 | 1.29× |
| **VLA-ADP (Ratio=50%)** | ✓ | PD | OFT (7B) | **99.4%** | 98.0% | 96.4% | 91.2% | **96.3%** | 6.43 | 1.23× |
| **VLA-ADP (Ratio=60%)** | ✓ | PD | OFT (7B) | 98.8% | 98.0% | 95.8% | **92.0%** | 96.2% | 6.74 | 1.17× |
| **VLA-ADP (Ratio=70%)** | ✓ | PD | OFT (7B) | 99.0% | 98.2% | 96.8% | 91.2% | **96.3%** | 7.03 | 1.13× |

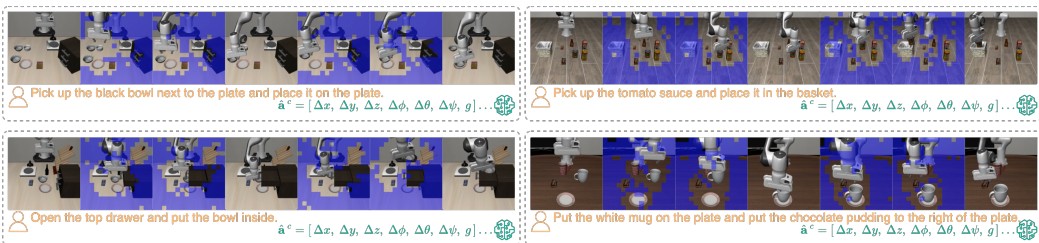

Figure 4: Visualisation of our method on representative examples of the four LIBERO task types (Spatial, Object, Goal, Long). Blue masks ■ indicate pruned vision tokens. The retained tokens consistently highlight task-relevant objects, validating the Text-driven Anticipatory Pruning. Moreover, full vision tokens are restored at critical phases (*e.g.*, initialisation, grasping, placement), demonstrating the effectiveness of the Task-driven Pruning and Action-aware Dynamic Strategy.

with expected savings

$$\mathbb{E}[\Delta \mathcal{F}_{\text{episode}}] = T \, \gamma \, \Delta F_{\text{ADP}}. \tag{25}$$

The pruning is applied at the embedding stage prior to $f_{\text{LLM}}$, so the reduced length $S'$ benefits all $H$ layers uniformly, and the dynamic rule (Eq. 14-16) controls how often the pruned path is used across action windows.

## 5 EXPERIMENTS

We conduct experiments across LIBERO simulation and real-robot tasks under standardized settings, comparing against strong baselines and conducting targeted ablations to assess success rates, compute/latency, and the contributions of dynamic scheduling and layer-wise pruning.

### 5.1 SIMULATION EXPERIMENTS

**Experiments setup.** For simulation, we use LIBERO Liu et al. (2023a) with four suites (*i.e.*, Spatial, Object, Goal, Long) evaluating spatial understanding, object recognition, goal-directed behaviour, and long-horizon planning. All LIBERO settings follow OpenVLA-OFT, using its public run scripts. For comparison, we reproduce FastV Chen et al. (2024) on OpenVLA-OFT and run under the same environment. Experiments run on Linux with an NVIDIA RTX 4090. We set the window size to the OpenVLA-OFT chunk size (8) and apply a cold start: the first two windows use full vision. To limit error accumulation, if pruning occurs in three consecutive windows, the next window is forced to full vision. In multi-view pruning, the main wrist retention is 4:6. Within the Action-aware Dynamic Strategy, we use Euclidean displacement and the adjacent-extrema rule; in its third case we deterministically set the state to 1 (instead of inheriting $s_i$) to further reduce FLOPs.

Table 2: **Real-World Experiments (4 tasks).** Task 1 to Task 4 cover the picking, placing and wiping motion, which covering a variety of objects.

| Method | Decoding | Task1 | Task2 | Task3 | Task4 | Average | Latency↓ | Speedup↑ |
|---|---|---|---|---|---|---|---|---|
| OpenVLA-OFT (base) | PD | 83.3% | **93.3%** | 86.7% | 80.0% | 85.8% | 76.9 | 1.00 |
| **VLA-ADP** (ours) | PD | **90.0%** | 90.0% | **90.0%** | **83.3%** | **88.3%** | **51.8** | **1.49×** |

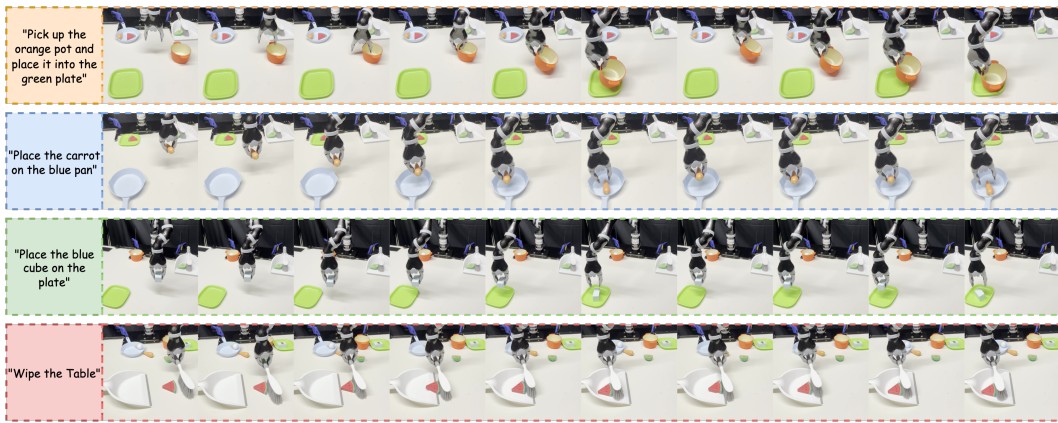

Figure 5: **Real world experiments**. We conduct the experiments on Jaco2 Real-world Platform.

**Main result.** Table 1 shows that VLA-ADP achieves a stable accuracy-compute trade-off across the keep ratio on LIBERO. Compared to OpenVLA-OFT, when the keep ratio is 50-70%, the average success rate slightly decreases ($\leq 0.9\%$), while the LLM-side inference speed improves by up to 1.23×, and the FLOPs are markedly below the baseline. Further compressing the retention rate to 30-40% maintains an average SR of 94.4-94.8%, yet attains 1.29-1.35× speedup. Notably, VLA-ADP achieves a 99.4% success rate on the Spatial, indicating that VLA-ADP successfully prunes redundant vision tokens while selectively preserving key information in relatively simple spatial manipulation scenarios. By contrast, Random Dropping (50%) yields a 1.29× speedup on the LLM side and performs reasonably on the Spatial and Goal, but its success rates on Object and Long are only 73.0% and 76.2%, respectively.

## 5.2 REAL-WORLD EXPERIMENTS

**Experiment setup.** We evaluate on four real tasks executed on a physical robot: *Task1*: pick up the orange pot and place it in the green plate; *Task2*: place the blue cube on the plate; *Task3*: place the carrot on the blue pan; *Task4*: wipe the table. All runtime settings mirror the simulation: Linux workstation with an NVIDIA RTX 4090, parallel decoding (PD) and window size equal to OpenVLA-OFT chunk size (8) with a two-window cold start (full vision). To curb error accumulation, if pruning is enabled for three consecutive windows, the next window uses full vision. Within the Action-aware Dynamic Strategy we use Euclidean displacement and the adjacent-extrema rule, and in the third case deterministically set the state to 1 to further reduce FLOPs.

**Main result.** Table 2 summarizes the outcomes. VLA-ADP improves the average success rate from 85.8% to 88.3% while reducing latency from 76.9 to 51.8 (baseline/ours = 1.49×). Per task, VLA-ADP outperforms the baseline on Task1/3/4 (90.0%, 90.0%, 83.3%) and is still competitive on Task2 (90.0% vs. 93.3%). These results indicate that our dynamic pruning maintains or improves real-world success while yielding substantial speedup.

## 6 ABLATION STUDY

We conduct ablations on each component of our method: first removing the dynamic strategy to test ADP (*w/o Action-aware Dynamic*); then ablating the pruning module (*w/o Text-driven Pruning*) with different weight selections for relevance scoring.

Table 3: **Ablation Study on each component of our design. (a)** Ablation on dynamic strategies ($f$). **(b)** Ablation on pruning based on weights retrieved from layer $l$. Metrics are SR (%) and FLOPs.

| $f$ | Spatial SR↑ | Object SR↑ | Goal SR↑ | Long SR↑ | Avg SR↑ | Avg $\rho^{avg}$ | FLOPs↓ | $l$ | Spatial SR↑ | Object SR↑ | Goal SR↑ | Long SR↑ | Avg SR↑ | Avg FLOPs↓ |
|---|---|---|---|---|---|---|---|---|---|---|---|---|---|---|
| ADP | **99.4%** | **98.0%** | **96.4%** | **91.2%** | **96.3%** | **0.22** | **6.43** | 0 | **99.4%** | **98.0%** | **96.4%** | **91.2%** | **96.3%** | **6.43** |
| - w/o D | 98.2% | 88.0% | 96.4% | 91.2% | 93.45% | 0.25 | 6.23 | 1 | 98.2% | 97.6% | 96.2% | 89.8% | 95.5% | 6.57 |
| - w/o D + PS | 95.0% | 81.4% | 96.2% | 87.0% | 89.9% | 0.50 | 4.55 | 4 | 98.0% | 97.8% | 96.4% | 91.2% | 95.8% | 6.89 |

(a) (b)

(a) Original Attention Maps    (b) Pruning Attention Maps.    (c) Vision tokens importance score.

Figure 6: The figure illustrates the layer-wise importance score of the vision tokens. (a) Vision-vision self-similarity (across layers) (b) Text→vision attention (after ADP) (c) Vision-token importance $\Phi$

**Action-aware Dynamic Strategy.** We assess the impact of the dynamic strategy with two baselines: removing the dynamic function (*ADP w/o D*) and a handcrafted periodic switching variant (*w/o D + PS*). Table 3a shows that introducing the dynamic controller (ADP) yields the best overall accuracy-compute balance: 96.3% average SR with $\rho^{avg} = 0.22$ and 6.43 FLOPs, outperforming the variant *w/o D* (93.45%, $\rho^{avg} = 0.25$, 6.23 FLOPs) and the periodic schedule *w/o D + PS* (89.9%, $\rho^{avg} = 0.50$, 4.55 FLOPs). Per-suite, the dynamic policy preserves Goal/Long while boosting Spatial/Object—most notably a **+16.6** point gain on *Object* over the periodic schedule (98.0% vs. 81.4%), and **+4.4** on *Spatial* (99.4% vs. 95.0%). Compared to *ADP w/o D*, ADP improves average SR by **+2.85** points with comparable compute (+0.20 FLOPs), indicating that state-aware switching (rather than fixed cycling) is crucial to avoid over-pruning during fine manipulation while still pruning aggressively when large motions occur.

**Impact of Pruning Strategy.** We ablate layer selection to validate using layer 0 for importance scoring. Table 3b shows that final SR is similar across layers, with layer 0 yielding the best accuracy-compute balance: 96.3% average SR with 6.43 FLOPs, compared to layer 1 (95.5%, 6.57 FLOPs) and layer 4 (95.8%, 6.89 FLOPs). Meanwhile, FLOPs increase with deeper layers, so selecting $l = 0$ offers a superior performance-efficiency trade-off. The slight decline in average SR with depth corroborates our analysis: deeper layers induce more localised attention—sharpening peaks vs. non-peaks but reducing global state coverage—thus becoming more sensitive to occasional mismatches and noise, which leads to marginally lower success rates.

**Observation of Attention Weights.** A common view holds that text-vision alignment concentrates in deeper multimodal layers, so high importance scores should be computed there; however, this need not hold for parallel-decoding VLA. Our visualizations and measurements show that layer 0 already provides a stable, discriminative text-to-vision signal. In Figure 6a, the layer-0 self-similarity matrix shows a clear high-contrast block structure, while deeper layers become increasingly diagonal-banded, compressing non-local correlations. The text to vision submatrix (Figure 6b) follows the same trend: layer 0 exhibits pronounced peaks and troughs across many vision tokens, whereas deeper layers retain only a few narrow high-response bands. The token importance score $\Phi$ (Figure 6c) likewise has higher SNR at layer 0; with depth, curves sharpen and develop long tails or near collapse, making top-$k$ ranking more sensitive to local noise.

# 7   CONCLUSION

We presented VLA-ADP, a plug-and-play pruning framework that unifies text-driven anticipatory pruning with an action-aware dynamic strategy to accelerate VLA inference while preserving reliabil-

ity. Across simulation and real-world evaluations, the method maintains or improves task success, reduces compute, and shortens inference latency. The dynamic controller consistently outperforms removal and periodic switching, and early-layer scoring offers the best accuracy-efficiency balance. These results indicate that adaptively pruning vision tokens by motion state and instruction relevance enables efficient, fine-grained manipulation without compromising control quality.

# 8 ACKNOWLEDGEMENTS

This work was supported in part by the Australian Research Council under Projects DP240101848 and FT230100549.

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

APPENDIX

THE USE OF LARGE LANGUAGE MODELS

In this manuscript, we used a LLM as a writing assistant to refine the presentation of certain contents. Specifically, the LLM was used to: (1) Rephrase sentences and paragraphs for greater readability, conciseness, and academic formality. (2) Correct grammar, spelling, and punctuation errors. (3) Improve logical flow and transitions between sentences. All content enhanced by the LLM has been thoroughly reviewed by the authors to ensure accuracy. The core idea, technical contributions, and results of the experiment are entirely the original contributions of the authors.

ALGORITHM

Here we provide the full pipeline of our proposed action-aware dynamic pruning for vision-language-action models.

---

**Algorithm 1** Action-aware Dynamic Pruning (ADP) for VLA Inference

---

**Require:** Multimodal embeddings $\mathbf{X}^m = \big[\mathbf{X}^{[\mathrm{BOS}]}, \mathbf{X}^{\mathrm{vis}}, \mathbf{X}^{\mathrm{prop}}, \mathbf{X}^{\mathrm{txt}}, \mathbf{X}^{\mathrm{act}}, \mathbf{X}^{[\mathrm{EOS}]}\big]$,

       Q/K weights $W_Q^{(0)}, W_K^{(0)}$, retention ratio $\rho$, view weights $\boldsymbol{\alpha} \in \mathbb{R}^C$ with $\sum_c \alpha_c = 1$,

       gating rule $f(\cdot)$ (mean or adjacent-extrema), windowed actions $\mathbf{A}_i^c$, pose $T_{b_i}$

**Output:** Per-window visual state $s_i \in \{0, 1\}$, pruned sequence $\tilde{\mathbf{X}}^m$, actions $\hat{\mathbf{a}}$

1:  **Init:** $s_1 \leftarrow 0$ (cold start, full vision), $s_2 \leftarrow 0$; $\texttt{consec} \leftarrow 0$             ▷ two-window cold start
2:  **for** window $i = 1, 2, \ldots$ **do**
3:      **Compute windowed motion** $\delta_i$ via FK on $\mathbf{A}_i^c$ (Def. 4.1):

$$\delta_i \leftarrow \sum_{t=b_i}^{e_i-1} \|\mathbf{p}_{t+1} - \mathbf{p}_t\|_2 \quad \text{with } \mathbf{p}_t = \pi(T_t),\ T_{t+1} = T_t \begin{bmatrix} R_{i,u} & \mathbf{v}_{i,u} \\ \mathbf{0}^\top & 1 \end{bmatrix}$$

4:      **Gate update:** $s_{i+1} \leftarrow f(\delta_i)$
5:      **if** $\texttt{consec} \geq 3$ **then**                  ▷ reset to avoid prolonged pruning
6:         $s_{i+1} \leftarrow 0$; $\texttt{consec} \leftarrow 0$
7:      **Branch by state**
8:      **if** $s_{i+1} = 1$ **then**                       ▷ pruned path
9:         **Q/K scoring at layer 0:**
           $\mathbf{Q}^{(0)} = \mathbf{H}_{\mathrm{txt}}^{(0)} W_Q^{(0)},\ \mathbf{K}^{(0)} = \mathbf{H}_{\mathrm{vis}}^{(0)} W_K^{(0)},\ \mathbf{A}^{(0)} = \frac{\mathbf{Q}^{(0)}(\mathbf{K}^{(0)})^\top}{\sqrt{d}}$
10:        **Aggregate importance** (Eq. 9): $\Phi^{(0)}(v) \leftarrow \frac{1}{N^h L_{\mathrm{txt}}} \sum_{h,t} \mathbf{A}_{h,t,v}^{(0)}$
11:        $k \leftarrow \lfloor \rho \cdot L^{\mathrm{vis}} \rfloor$; $k_c \leftarrow \lfloor \alpha_c \cdot k \rfloor$ for $c = 1..C$
12:        **Per-view Top-$k$:** $\mathbf{X}_{(c)}^{\mathrm{vis}} \leftarrow \text{Top-K}(\Phi_{(c)}^{(0)}, k_c)$; $\mathbf{X}_{\mathrm{keep}}^{\mathrm{vis}} \leftarrow \bigcup_c \mathbf{X}_{(c)}^{\mathrm{vis}}$
13:        **Form pruned input:** $\tilde{\mathbf{X}}^m \leftarrow [\mathbf{X}^{[\mathrm{BOS}]}, \mathbf{X}_{\mathrm{keep}}^{\mathrm{vis}}, \mathbf{X}^{\mathrm{prop}}, \mathbf{X}^{\mathrm{txt}}, \mathbf{X}^{\mathrm{act}}, \mathbf{X}^{[\mathrm{EOS}]}]$
14:        **LLM forward:** $\hat{\mathbf{a}} \leftarrow f_{\mathrm{LLM}}(\tilde{\mathbf{X}}^m)$; $\texttt{consec} \leftarrow \texttt{consec} + 1$
15:      **else**                                  ▷ full-vision path
16:        $\tilde{\mathbf{X}}^m \leftarrow \mathbf{X}^m$; $\hat{\mathbf{a}} \leftarrow f_{\mathrm{LLM}}(\mathbf{X}^m)$; $\texttt{consec} \leftarrow 0$
17:      **Execute** $\hat{\mathbf{a}}$ and advance to next window
18: **return** $\{s_i\}, \tilde{\mathbf{X}}^m, \hat{\mathbf{a}}$

---

REAL-WORLD EXPERIMENT DETAILS

We provide additional details of the real-robot experiments to facilitate reproducibility.

**Robot Platform.** All real-world evaluations were conducted on a **Kinova Jaco2** 6-DoF robotic arm equipped with a parallel-jaw gripper. The robot was controlled through Cartesian velocity commands at 10 Hz, with joint and workspace safety limits enforced to prevent collisions.

**Sensing Setup.** A single RGB camera (Sony AX53) was mounted in front of the table to capture the entire workspace. The camera streamed RGB frames at $640 \times 480$ resolution and 30 FPS. No wrist-mounted camera was used in the real-robot experiments, and all observations were obtained from this fixed view.

**Runtime Environment.** All policies were executed on a Linux workstation running Ubuntu 20.04 with an NVIDIA RTX 4090 GPU. Inference employed parallel decoding with an OFT chunk size of 8. Latency was measured as mean per-step inference time over 100 runs.

**Task Setup.** We evaluated four tabletop manipulation tasks: **a) Task 1:** Pick up the orange pot and place it into a green plate. **b) Task 2:** Pick and place a blue cube onto the plate. **c) Task 3:** Pick and place a carrot into a blue pan. **d) Task 4:** Wipe the table. Objects were placed randomly within a $50 \times 50$ cm workspace area at the beginning of each trial. For data collection, each task was executed using a gamepad teleoperation interface, with the human operator directly controlling the Jaco2 arm. We collected approximately 100–150 trajectories per task, covering diverse initial states and object positions. The collected demonstrations were then used to build the training dataset, and we fine-tuned OpenVLA-OFT separately on each task before deploying the model for evaluation in the real-robot experiments.

**Evaluation Protocol.** Each task was repeated 30 trials under randomized object initializations. A trial was marked as successful if the object was placed entirely within the target receptacle (for Tasks 1–3) or if more than $80\%$ of the designated area was wiped (Task 4). At the end of each episode, the robot was reset to a neutral home pose. Safety termination was triggered if joint torque exceeded preset thresholds.

QUANTITATIVE STATISTICS

To complement the attention visualizations with comparable quantitative evidence, we compute scale-free statistics at each layer from the text-to-vision importance vector $\Phi = \{\phi_i\}_{i=1}^{V}$. We first normalize $\Phi$ as $p_i = \phi_i / \sum_{j=1}^{V} \phi_j$ and then evaluate the *Participation Ratio (PR)* and the *Entropy ($\mathcal{H}$)*:

$$PR = \frac{1}{\sum_{i=1}^{V} p_i^2}, \quad \mathcal{H}(p) = -\sum_{i=1}^{V} p_i \log p_i. \tag{26}$$

$PR$ approximates the effective number of participating visual columns, while $\mathcal{H}$ quantifies the dispersion and multi-modality of the distribution. Together, they characterize the spatial coverage of alignment signals and the stability of Top-$K$ ranking.

Figure 7a presents the variation of *PR* across layers. In shallow layers, *PR* remains relatively high, indicating that a large number of key vision tokens receive balanced attention and contribute to broad spatial coverage. As depth increases, *PR* rapidly decreases and stabilizes, reflecting a progressive concentration of attention onto a small subset of tokens and reduced feature utilization.

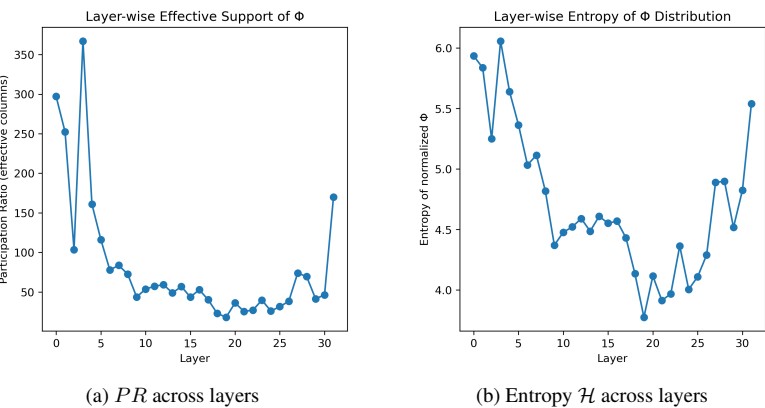

(a) $PR$ across layers

(b) Entropy $\mathcal{H}$ across layers

Figure 7: Layer-wise statistics of text-to-vision importance scores.

Figure 7b presents the corresponding entropy $\mathcal{H}$ across layers. Shallow layers exhibit higher entropy, suggesting diverse and smooth distributions that make Top-$K$ selection robust against individual outliers. In contrast, deeper layers display sharply peaked distributions dominated by a few tokens. While such localization may benefit semantic alignment in downstream reasoning, it makes Top-$K$ selection unstable and prone to discarding critical tokens. These findings suggest that shallow-layer attention signals are more suitable for estimating the importance of vision tokens in VLA tasks. In particular, using layer-0 offers a computationally efficient choice, providing reliable importance estimates while further reducing FLOPs.

RANDOM PRUNING COMPARISON

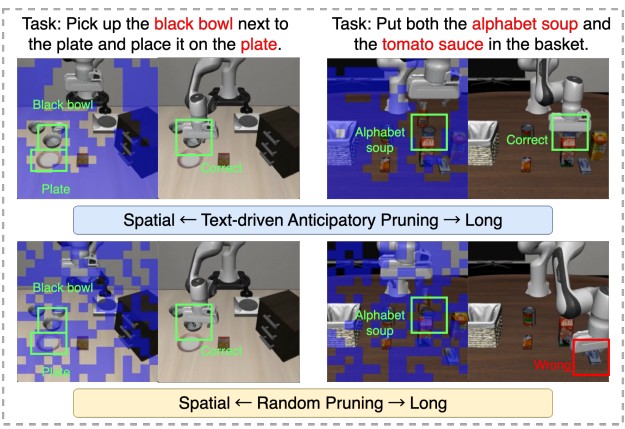

Figure 8: ADP (Text-driven Pruning) vs. Random Pruning.

Random Pruning yields competitive performance on Spatial and Goal, but performs substantially worse on Object and Long. We can attribute this phenomenon to the distribution of visual patches spanned by the key object as shown in Figure 8. When the target object spans a large number of patches, random pruning still has a high probability of retaining the relevant patches, thereby preserving sufficient task-critical vision information. In contrast, random pruning is more likely to prune the relevant patches when the key object spans only a small number of patches. Such pruning may result in incorrect object identification, which in turn propagates errors and ultimately causes task failure.

INVERSION EXPERIMENT

Table 4: Inversion Experiment.

| Task | Keep Ratio=30% | | | Keep Ratio=50% | | | Keep Ratio=70% | | |
|---|---|---|---|---|---|---|---|---|---|
| | Direct | Inverse | Gap | Direct | Inverse | Gap | Direct | Inverse | Gap |
| Spatial | 97.6% | 44.0% | 53.6% | 99.4% | 54.8% | 44.6% | 99.0% | 90.8% | 8.2% |
| Object | 98.4% | 0.6% | 97.8% | 98.0% | 11.8% | 86.2% | 98.2% | 59.6% | 38.6% |
| Goal | 97.4% | 43.6% | 53.8% | 96.4% | 63.6% | 32.8% | 96.8% | 84.8% | 12.0% |
| Long | 84.2% | 0.2% | 84.0% | 91.2% | 8.2% | 83.0% | 91.2% | 37.4% | 53.8% |
| Avg | 94.4% | 22.1% | 72.3% | 96.3% | 34.6% | 61.7% | 96.3% | 68.1% | 28.1% |

To further evaluate the reliability of our text-driven importance scores, we design an inverse-pruning experiment that uses the same experimental setup as the main results. Instead of retaining the most relevant visual tokens identified by our text-vision relevance ranking, we intentionally preserve the least relevant ones, while keeping all other factors, including model, hyperparameters, tasks, and ratios (30%, 50%, 70%), unchanged. If the relevance ranking were noisy or poorly aligned with task semantics, the performance gap between direct and inverse pruning would be small. Conversely,

a large and systematic divergence would indicate that the scoring mechanism effectively captures task-relevant visual information.

The results in Table 4 show a clear and consistent performance gap between the two settings across all suites and keep ratios. Under a 30% keep ratio, the average success rate drops sharply from 94.4% (direct) to 22.1% (inverse), with near-complete failure on Object (0.6%) and Long (0.2%). Similar trends persist at 50% (from 96.3% to 34.6%) and 70% (from 96.3% to 68.1%). Notably, inverse-70% still performs far worse than direct-30%, highlighting that semantic relevance dominates task performance over the number of retained tokens.

Notably, the degradation is particularly pronounced on Object and Long, which require fine-grained spatial reasoning and long-horizon planning. Under inverse pruning, the model fails to preserve the high-ranked, task-relevant visual tokens. This provides strong evidence that our importance scores do not drift toward irrelevant regions but consistently identify and prioritise semantically aligned visual tokens. In summary, these counterfactual results further support the conclusion that:

1. Task-relevant visual tokens play a decisive role in VLA tasks, and task success depends far more on preserving these semantically aligned tokens than on the number of tokens retained. Task performance degrades sharply as soon as these high-relevance tokens are removed, demonstrating that semantic alignment is the dominant factor influencing model performance.

2. The text-driven relevance scoring mechanism effectively identifies the task-relevant visual tokens, showing that ADP's anticipatory pruning relies on a stable and meaningful semantic signal rather than incidental attention fluctuations. The scoring module effectively captures the alignment between the task instruction and the image observations, identifying the visual patches critical for object grounding, spatial reasoning, and contact-rich manipulation.

COMPUTATIONAL TRADE-OFF ANALYSIS

Table 5: FLOPs and speedup with $0.3$ ratio. ADP($l$) denotes pruning based on the weights from the i-th layer.

| Item | baseline | ADP (0) | ADP (2) | ADP (4) | ADP (8) | ADP (16) | ADP (24) |
|---|---|---|---|---|---|---|---|
| Keep Ratio | 1.0 | 0.3 | 0.3 | 0.3 | 0.3 | 0.3 | 0.3 |
| Scoring Layer | – | 0 | 2 | 4 | 8 | 16 | 24 |
| Seq. Len. | 603 | 245 | 245 | 245 | 245 | 245 | 245 |
| Vis. Tokens (kept) | 512 | 154 | 154 | 154 | 154 | 154 | 154 |
| Scoring FLOPs (T) | 0.00 | 0.02 | 0.51 | 1.01 | 1.99 | 3.97 | 5.95 |
| LLM FLOPs (T) | 7.91 | 3.19 | 3.19 | 3.19 | 3.19 | 3.19 | 3.19 |
| Total FLOPs (T) | 7.91 | 3.21 | 3.70 | 4.20 | 5.18 | 7.16 | 9.14 |
| Expected FLOPs (T) | 7.91 | 5.85 | 6.06 | 6.27 | 6.71 | 7.58 | 8.45 |
| Speedup (Orig/Exp.) | 1.00 | 1.35 | 1.31 | 1.26 | 1.18 | 1.04 | 0.94 |

Table 6: FLOPs and speedup with keep ratio $0.4$.

| Item | baseline | ADP (0) | ADP (2) | ADP (4) | ADP (8) | ADP (16) | ADP (24) |
|---|---|---|---|---|---|---|---|
| Keep Ratio | 1.0 | 0.4 | 0.4 | 0.4 | 0.4 | 0.4 | 0.4 |
| Scoring Layer | – | 0 | 2 | 4 | 8 | 16 | 24 |
| Seq. Len. | 603 | 296 | 296 | 296 | 296 | 296 | 296 |
| Vis. Tokens (kept) | 512 | 205 | 205 | 205 | 205 | 205 | 205 |
| Scoring FLOPs (T) | 0.00 | 0.02 | 0.51 | 1.01 | 1.99 | 3.97 | 5.95 |
| LLM FLOPs (T) | 7.91 | 3.86 | 3.86 | 3.86 | 3.86 | 3.86 | 3.86 |
| Total FLOPs (T) | 7.91 | 3.88 | 4.37 | 4.86 | 5.85 | 7.83 | 9.80 |
| Expected FLOPs (T) | 7.91 | 6.14 | 6.35 | 6.57 | 7.00 | 7.87 | 8.74 |
| Speedup (Orig/Exp.) | 1.00 | 1.29 | 1.25 | 1.20 | 1.13 | 1.00 | 0.90 |

To quantify the computational trade-off introduced by anticipatory pruning, we evaluate the FLOPs associated with importance scoring and compare them against the savings obtained from reducing the

Table 7: FLOPs and speedup with keep ratio 0.5.

| Item | baseline | ADP (0) | ADP (2) | ADP (4) | ADP (8) | ADP (16) | ADP (24) |
|---|---|---|---|---|---|---|---|
| Keep Ratio | 1.0 | 0.5 | 0.5 | 0.5 | 0.5 | 0.5 | 0.5 |
| Scoring Layer | – | 0 | 2 | 4 | 8 | 16 | 24 |
| Seq. Len. | 603 | 347 | 347 | 347 | 347 | 347 | 347 |
| Vis. Tokens (kept) | 512 | 256 | 256 | 256 | 256 | 256 | 256 |
| Scoring FLOPs (T) | 0.00 | 0.02 | 0.51 | 1.01 | 1.99 | 3.97 | 5.95 |
| LLM FLOPs (T) | 7.91 | 4.53 | 4.53 | 4.53 | 4.53 | 4.53 | 4.53 |
| Total FLOPs (T) | 7.91 | 4.55 | 5.04 | 5.53 | 6.52 | 8.50 | 10.47 |
| Expected FLOPs (T) | 7.91 | 6.43 | 6.64 | 6.89 | 7.30 | 8.17 | 9.04 |
| Speedup (Orig/Exp.) | 1.00 | 1.23 | 1.19 | 1.15 | 1.08 | 0.97 | 0.87 |

Table 8: FLOPs and speedup with 0.6 ratio.

| Item | baseline | ADP (0) | ADP (2) | ADP (4) | ADP (8) | ADP (16) | ADP (24) |
|---|---|---|---|---|---|---|---|
| Keep Ratio | 1.0 | 0.6 | 0.6 | 0.6 | 0.6 | 0.6 | 0.6 |
| Scoring Layer | – | 0 | 2 | 4 | 8 | 16 | 24 |
| Seq. Len. | 603 | 399 | 399 | 399 | 399 | 399 | 399 |
| Vis. Tokens (kept) | 512 | 308 | 308 | 308 | 308 | 308 | 308 |
| Scoring FLOPs (T) | 0.00 | 0.02 | 0.51 | 1.01 | 1.99 | 3.97 | 5.95 |
| LLM FLOPs (T) | 7.91 | 5.21 | 5.21 | 5.21 | 5.21 | 5.21 | 5.21 |
| Total FLOPs (T) | 7.91 | 5.23 | 5.72 | 6.22 | 7.20 | 9.18 | 11.16 |
| Expected FLOPs (T) | 7.91 | 6.74 | 6.94 | 7.16 | 7.60 | 8.47 | 9.34 |
| Speedup (Orig/Exp.) | 1.00 | 1.17 | 1.14 | 1.10 | 1.04 | 0.93 | 0.85 |

**LLM sequence length.** Tables 5-9 report detailed FLOPs measurements across a range of keep ratios and scoring layers (all FLOPs reported in this section are computed under the original multimodal sequence length of $S = 603$.)

Across all practically meaningful configurations, the scoring cost remains substantially smaller than the compute saved by pruning. When importance ranking is performed using shallow layers (e.g. layers 0-4), the scoring overhead remains within 0.02-1.01T FLOPs. In contrast, shortening the visual sequence reduces the LLM compute from 7.91T to approximately 3.19-5.90T, yielding a 2-4T reduction. The net effect is therefore dominated by the savings from operating on a shorter sequence, and the overall inference cost decreases consistently.

When combined with the action-aware dynamic gating strategy, the expected inference cost reflects the mixture of pruned and full-vision windows. In our implementation, the dynamic gate activates pruning for roughly 44% of windows on average across LIBERO. Under this configuration, the expected FLOPs per forward are 5.85T (keep=0.3), 6.14T (keep=0.4), and 6.43T (keep=0.5), corresponding to $1.35\times$, $1.29\times$, and $1.23\times$ speedup over the baseline. These values already account for the scoring overhead, confirming that the computational benefit persists under realistic usage.

We also observe that performing relevance scoring at excessively deep layers ($\geq 16$) increases the overhead to 3.97-5.95T FLOPs, which approaches or surpasses the FLOPs saved by pruning. In such cases, the net speedup degrades and may even become slightly negative. However, such configurations are not required in practice. Our ablation study already shows that layer-0 scoring not only minimises computational overhead but also provides the most reliable and stable importance estimates, making it the default choice in all experiments.

Overall, the additional scoring computation is far smaller than the savings achieved through reduced sequence length for all practical settings, and the use of shallow-layer scoring ensures that the trade-off remains consistently positive. These results demonstrate that the proposed pruning strategy offers genuine computational benefits without compromising performance.

Table 9: FLOPs and speedup with keep ratio 0.7.

| Item | baseline | ADP (0) | ADP (2) | ADP (4) | ADP (8) | ADP (16) | ADP (24) |
|---|---|---|---|---|---|---|---|
| Keep Ratio | 1.0 | 0.7 | 0.7 | 0.7 | 0.7 | 0.7 | 0.7 |
| Scoring Layer | – | 0 | 2 | 4 | 8 | 16 | 24 |
| Seq. Len. | 603 | 450 | 450 | 450 | 450 | 450 | 450 |
| Vis. Tokens (kept) | 512 | 359 | 359 | 359 | 359 | 359 | 359 |
| Scoring FLOPs (T) | 0.00 | 0.02 | 0.51 | 1.01 | 1.99 | 3.97 | 5.95 |
| LLM FLOPs (T) | 7.91 | 5.88 | 5.88 | 5.88 | 5.88 | 5.88 | 5.88 |
| Total FLOPs (T) | 7.91 | 5.90 | 6.39 | 6.89 | 7.88 | 9.85 | 11.83 |
| Expected FLOPs (T) | 7.91 | 7.03 | 7.24 | 7.46 | 7.89 | 8.76 | 9.63 |
| Speedup (Orig/Exp.) | 1.00 | 1.13 | 1.09 | 1.06 | 1.00 | 0.90 | 0.82 |

CASE DISCUSSIONS

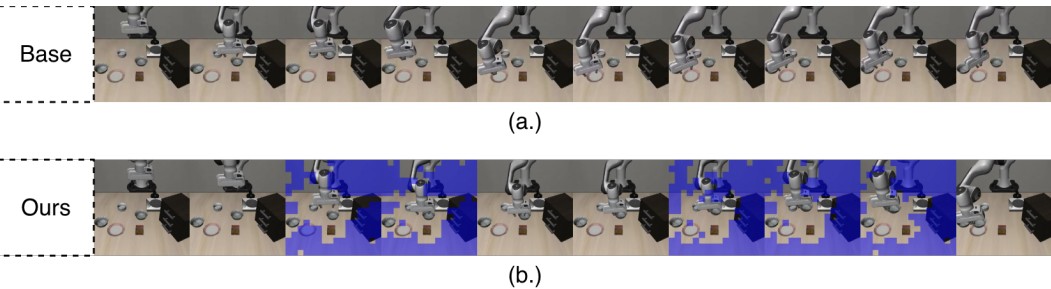

(a.)

(b.)

Figure 9: Baseline Failed but Ours Success on *#Episode 110* of Spatial task.

**Performance Improvements through Pruning.** We observe an intriguing phenomenon: in several scenarios, the pruned model performs even better than the original baseline. To better understand this behavior, we further examine representative cases and analyze the underlying factors that contribute to such improvements. We identify a representative case that illustrates why moderate pruning can outperform the unpruned baseline. In Figure 9a, the baseline model fails to grasp the bowl, whereas in Figure 9b our ADP method succeeds with a stable grasp. A closer inspection shows that the baseline allocates non-trivial attention to large, static background surfaces, effectively behaving like "sink" regions that dilute focus. After pruning these low-value background tokens, the model concentrates its attention on the immediate manipulation region—specifically the bowl and its supporting surface. This reduction of peripheral noise results in a more discriminative and task-aligned visual representation, helping the policy make more precise action decisions. Thus, in cases where background clutter dominates the unpruned attention distribution, pruning can improve operational focus and lead to higher success rates than the original baseline.

**Limitations in Long-Horizon Tasks.** In addition to the individual examples, we analyze the per-task performance gap between ADP (keep ratio 0.5) and the OpenVLA-OFT baseline across all LIBERO suites (Tables 10-13). The performance drop is almost exclusively concentrated in the Long suite, while the Spatial, Object, and Goal suites remain largely unaffected, with comparable or slightly improved success rates. Notably, the largest degradations occur in tasks with long, compositional instructions involving multiple sequential subtasks, where the model must progressively shift visual attention across different stages and object sets.

This phenomenon highlights a limitation of current VLA backbones. Since models such as OpenVLA-OFT generate actions based only on the current multimodal context and a static instruction, they lack explicit awareness of subgoal completion. Consequently, visual tokens associated with earlier subtasks may still be retained due to textual relevance, even after the subgoal has been achieved. Under constrained token budgets, this misallocation becomes amplified, leaving insufficient visual coverage for subsequent stages and leading to failure. Our inversion analysis further confirms that VLA performance is highly sensitive to the availability of correct visual tokens.

Table 10: Spatial Suite Performance (ADP vs Baseline).

| Task | Length | ADP | Baseline | Gap |
|---|---|---|---|---|
| pick up the black bowl between the plate and the ramekin and place it on the plate | 82 | 100.0% | 100.0% | 0.0% |
| pick up the black bowl next to the ramekin and place it on the plate | 68 | 100.0% | 100.0% | 0.0% |
| pick up the black bowl from table center and place it on the plate | 66 | 100.0% | 100.0% | 0.0% |
| pick up the black bowl on the cookie box and place it on the plate | 66 | 100.0% | 98.0% | +2.0% |
| pick up the black bowl in the top drawer of the wooden cabinet and place it on the plate | 88 | 98.0% | 94.0% | +4.0% |
| pick up the black bowl on the ramekin and place it on the plate | 63 | 98.0% | 100.0% | -2.0% |
| pick up the black bowl next to the cookie box and place it on the plate | 71 | 100.0% | 100.0% | 0.0% |
| pick up the black bowl on the stove and place it on the plate | 61 | 98.0% | 98.0% | 0.0% |
| pick up the black bowl next to the plate and place it on the plate | 66 | 100.0% | 100.0% | 0.0% |
| pick up the black bowl on the wooden cabinet and place it on the plate | 70 | 100.0% | 96.0% | +4.0% |

Table 11: Object Suite Performance (ADP vs Baseline).

| Task | Length | ADP | Baseline | Gap |
|---|---|---|---|---|
| pick up the alphabet soup and place it in the basket | 52 | 100.0% | 98.0% | +2.0% |
| pick up the cream cheese and place it in the basket | 51 | 100.0% | 100.0% | 0.0% |
| pick up the salad dressing and place it in the basket | 53 | 98.0% | 98.0% | 0.0% |
| pick up the bbq sauce and place it in the basket | 48 | 98.0% | 100.0% | -2.0% |
| pick up the ketchup and place it in the basket | 46 | 98.0% | 100.0% | -2.0% |
| pick up the tomato sauce and place it in the basket | 51 | 100.0% | 100.0% | 0.0% |
| pick up the butter and place it in the basket | 45 | 96.0% | 98.0% | -2.0% |
| pick up the milk and place it in the basket | 43 | 98.0% | 96.0% | +2.0% |
| pick up the chocolate pudding and place it in the basket | 56 | 94.0% | 92.0% | +2.0% |
| pick up the orange juice and place it in the basket | 51 | 98.0% | 100.0% | -2.0% |

While previous multimodal works Xu et al. (2025a); Wang et al. (2024); Su et al. (2025) have explored token pruning and efficiency trade-offs, these observations highlight that, in VLA settings, long-horizon compositional tasks introduce additional challenges related to subgoal progression and stage-aware attention reallocation. Future improvements may therefore require explicit mechanisms for subgoal estimation or dynamic visual attention redistribution.

IMPLEMENTATION DETAILS

**Hyperparameters.**   We provide the full set of hyperparameters used in our main experiments for reproducibility. These include all query-key (QK) pruning settings, dynamic gating parameters, and visualization options. Table 14 summarizes the exact values used in all results reported in the main text.

Table 12: Goal Suite Performance (ADP vs Baseline).

| Task | Length | ADP | Baseline | Gap |
|------|--------|-----|----------|-----|
| open the middle drawer of the cabinet | 37 | 100.0% | 98.0% | +2.0% |
| put the bowl on the stove | 25 | 94.0% | 96.0% | -2.0% |
| put the wine bottle on top of the cabinet | 41 | 94.0% | 92.0% | +2.0% |
| open the top drawer and put the bowl inside | 43 | 82.0% | 88.0% | -6.0% |
| put the bowl on top of the cabinet | 34 | 100.0% | 100.0% | 0.0% |
| push the plate to the front of the stove | 40 | 98.0% | 100.0% | -2.0% |
| put the cream cheese in the bowl | 32 | 98.0% | 96.0% | +2.0% |
| turn on the stove | 17 | 100.0% | 100.0% | 0.0% |
| put the bowl on the plate | 25 | 98.0% | 98.0% | 0.0% |
| put the wine bottle on the rack | 31 | 100.0% | 98.0% | +2.0% |

Table 13: Long Suite Performance (ADP vs Baseline).

| Task | Length | ADP | Baseline | Gap |
|------|--------|-----|----------|-----|
| put both the alphabet soup and the tomato sauce in the basket | 61 | 94.0% | 92.0% | +2.0% |
| put both the cream cheese box and the butter in the basket | 58 | 96.0% | 96.0% | 0.0% |
| turn on the stove and put the moka pot on it | 44 | 100.0% | 98.0% | +2.0% |
| put the black bowl in the bottom drawer of the cabinet and close it | 67 | 96.0% | 94.0% | +2.0% |
| put the white mug on the left plate and put the yellow and white mug on the right plate | 87 | 70.0% | 100.0% | -30.0% |
| pick up the book and place it in the back compartment of the caddy | 66 | 92.0% | 100.0% | -8.0% |
| put the white mug on the plate and put the chocolate pudding to the right of the plate | 86 | 80.0% | 98.0% | -18.0% |
| put both the alphabet soup and the cream cheese box in the basket | 65 | 98.0% | 98.0% | 0.0% |
| put both moka pots on the stove | 31 | 92.0% | 80.0% | +12.0% |
| put the yellow and white mug in the microwave and close it | 58 | 94.0% | 92.0% | +2.0% |

**Code Availability.** Our code and complete experimental configuration are publicly available at the project page. The implementation exposes all gating and pruning parameters through a unified configuration interface, making it easy to reproduce and extend the experiments (e.g., varying window sizes, pruning ratios, and similarity layers).

Table 14: Complete hyperparameter configuration used in all main experiments.

| QK-related Param | Value | Dynamic Param | Value | Viz/Eval Param | Value |
|------------------|-------|---------------|-------|----------------|-------|
| qk_keep_enabled | true | use_dynamic_visual_strategy | true | num_trials_per_task | 50 |
| qk_layer | 0 | decision_method | adjacent | overlay_pruned_color | [0,0,255] |
| qk_keep_ratio | 0.5 | adjacent_variant | extrema | overlay_pruned_alpha | 120 |
| qk_keep_split | [0.4, 0.6] | adjacent_extrema_window | 3 | video_fps | 10 |
| random_keep_ratio | 0.5 | adjacent_lookback | 2 | | |
| random_state_prob | 0.5 | delta_method | net | | |
| random_seed | 42 | limit_consecutive_pruned_enabled | true | | |
| | | limit_max_consecutive_pruned | 3 | | |

