# OpenReview forum: "Action-aware Dynamic Pruning for Efficient Vision-Language-Action Manipulation"
_ICLR.cc/2026/Conference — ICLR 2026 Poster_

### Official Review · Reviewer_4B1i · 2025-10-28

**Soundness:** 3
**Presentation:** 3
**Contribution:** 2
**Rating:** 4
**Confidence:** 4

**Summary:**

The paper addresses the inference efficiency problem of Vision-Language-Action (VLA) models in robotic manipulation and proposes a method called Action-Aware Dynamic Pruning (ADP). The core observation is that the redundancy of visual tokens is not constant during long-horizon manipulation tasks—it varies across different operation stages: redundancy is higher during coarse movements, while fine-grained grasping requires more detailed visual information. Based on this insight, the authors design two mechanisms to tackle the issue.

**Strengths:**

1. The method is clearly structured, and the writing makes it easy for readers to follow.

2. The experimental evaluation is relatively comprehensive.

**Weaknesses:**

1. I still believe that the core contribution (i.e., gating + pruning) demonstrates only a moderate level of novelty since it is relatively intuitive rather than a fundamentally transformative architectural innovation.

**Questions:**

1. In Figure 1, the word “pick” is mistakenly written as “picke.”

2. Although the paper discusses existing VLA acceleration methods (such as attention token pruning and structured compression), I think the authors should more clearly explain how ADP differs from these approaches in terms of innovation and performance boundaries.

3. I am also curious about the trade-off: since the paper introduces additional modules to learn how to prune tokens, at what level of token reduction does the time saved roughly offset the extra computation introduced by these modules?

4. From my previous experience, using text to attend to visual features often results in attention being paid to irrelevant regions. Does your method address this issue?

6. The authors propose to determine whether the current stage is “coarse” or “fine” based on recent changes in the end-effector’s trajectory. However, is this mechanism generalizable? Does it require manually set thresholds? And when the task type changes (e.g., when the motion involves mostly rotation rather than grasping), could this mechanism fail?

---

> ### Author Response · Authors · 2025-11-23
>
> **Re W1 && Q2: Novelty and Difference**
>
> We greatly appreciate your candid perspective on the novelty of our contribution. While ADP does not aim to introduce a fundamentally new architecture, its core innovation lies in bringing a new *action-aware* dimension into the VLA acceleration direction.
>
> Existing acceleration methods typically focus on (1) **static visual token pruning or compression**, where tokens are removed based on fixed heuristics or training-time sparsification, or (2) **architecture-level redesign**, such as lightweight backbones, layer skipping, or reparameterized blocks. These approaches implicitly assume that visual redundancy is *time-invariant* and do not model how the relevance of visual tokens evolves across manipulation stages.
>
> In contrast, ADP explicitly leverages the dynamics of the end-effector to detect phase transitions between coarse and fine-grained manipulation, and integrates this with text-driven semantic scoring to produce **action-aware, multimodal, dynamic pruning at inference time**. This allows the model to prune aggressively during high-motion phases while automatically preserving full visual fidelity during delicate operations. To our knowledge, no prior VLA acceleration method uses action dynamics as an inference-time decision variable for token retention.
>
> ADP therefore occupies a different point in the design space: it is not a replacement for structured compression or architectural redesign, but rather a **training-free, plug-and-play inference mechanism** that complements existing techniques and offers clear benefits in multi-phase manipulation scenarios where visual redundancy changes over time.
>
> We appreciate your thoughtful comment and the opportunity to clarify the positioning of our contribution.
>
>
>
> **Re Q1: Typo in Figure 1**
>
> We appreciate your careful observation regarding the typo in Figure 1. We have made the corrections, and we appreciate your careful review of our manuscript.

---

> ### Author Response · Authors · 2025-11-23
>
> **Re Q3: Token-Reduction Trade-off**
>
> We sincerely thank you for bringing this important question to our attention. We fully agree that any additional scoring overhead must remain substantially lower than the FLOPs saved by reducing the LLM sequence length. To quantify this, we conducted a detailed FLOPs calculation across different keep ratios and scoring layers, and we summarise our findings below.
>
> Across all practically meaningful configurations, the scoring cost is far smaller than the savings from pruning. For instance, when importance scoring is performed using shallow layers (e.g., layer 0–4), the scoring overhead stays within **0.02–1.01T FLOPs**, whereas pruning reduces the LLM compute from **7.91T** to roughly **3.19–5.90T**, producing a **2–4T reduction**. This means the savings from shortening the sequence dominate the additional scoring cost, and the net computational gain remains significant.
>
> When integrated with our dynamic gating strategy, in our implementation, the dynamic gating prunes approximately 44% of windows on average (measured on LIBERO), the expected inference cost per forward becomes 5.85T (keep=0.3), 6.13T (keep=0.4), and 6.43T (keep=0.5), corresponding to **1.35×, 1.29×, and 1.23× speedup** over the baseline. These values already account for both pruned and unpruned windows, confirming that the method provides consistent acceleration in realistic settings.
>
>
> | Method | Keep Ratio | Scoring Layer | Sequence Length after Pruning | Kept Vision Tokens | Scoring FLOPs (T) | LLM FLOPs on Pruned Sequence (T) | Total FLOPs per Pruned Forward (T) | Expected FLOPs per Forward (T) | Speedup (Orig/Expected) |
> |--------|------------|----------------|-------------------------------|---------------------|--------------------|----------------------------------|-------------------------------------|-------------------------------|--------------------------|
> | baseline | 1 |  | 603 | 512 | 0 | 7.91 | 7.91 | 7.91 | 1 |
> | ADP | 0.3 | 0 | 245 | 154 | 0.02 | 3.19 | 3.21 | 5.85 | 1.35 |
> | ADP | 0.3 | 2 | 245 | 154 | 0.51 | 3.19 | 3.7 | 6.06 | 1.31 |
> | ADP | 0.3 | 4 | 245 | 154 | 1.01 | 3.19 | 4.2 | 6.27 | 1.26 |
> | ADP | 0.3 | 8 | 245 | 154 | 1.99 | 3.19 | 5.18 | 6.71 | 1.18 |
> | ADP | 0.3 | 16 | 245 | 154 | 3.97 | 3.19 | 7.16 | 7.58 | 1.04 |
> | ADP | 0.3 | 24 | 245 | 154 | 5.95 | 3.19 | 9.14 | 8.45 | 0.94 |
> | ADP | 0.4 | 0 | 296 | 205 | 0.02 | 3.86 | 3.88 | 6.14 | 1.29 |
> | ADP | 0.4 | 2 | 296 | 205 | 0.51 | 3.86 | 4.37 | 6.35 | 1.25 |
> | ADP | 0.4 | 4 | 296 | 205 | 1.01 | 3.86 | 4.86 | 6.57 | 1.2 |
> | ADP | 0.4 | 8 | 296 | 205 | 1.99 | 3.86 | 5.85 | 7 | 1.13 |
> | ADP | 0.4 | 16 | 296 | 205 | 3.97 | 3.86 | 7.83 | 7.87 | 1 |
> | ADP | 0.4 | 24 | 296 | 205 | 5.95 | 3.86 | 9.8 | 8.74 | 0.9 |
> | ADP | 0.5 | 0 | 347 | 256 | 0.02 | 4.53 | 4.54 | 6.43 | 1.23 |
> | ADP | 0.5 | 2 | 347 | 256 | 0.51 | 4.53 | 5.04 | 6.64 | 1.19 |
> | ADP | 0.5 | 4 | 347 | 256 | 1.01 | 4.53 | 5.53 | 6.89 | 1.15 |
> | ADP | 0.5 | 8 | 347 | 256 | 1.99 | 4.53 | 6.52 | 7.3 | 1.08 |
> | ADP | 0.5 | 16 | 347 | 256 | 3.97 | 4.53 | 8.5 | 8.17 | 0.97 |
> | ADP | 0.5 | 24 | 347 | 256 | 5.95 | 4.53 | 10.47 | 9.04 | 0.87 |
> | ADP | 0.6 | 0 | 399 | 308 | 0.02 | 5.21 | 5.23 | 6.74 | 1.17 |
> | ADP | 0.6 | 2 | 399 | 308 | 0.51 | 5.21 | 5.72 | 6.94 | 1.14 |
> | ADP | 0.6 | 4 | 399 | 308 | 1.01 | 5.21 | 6.22 | 7.16 | 1.1 |
> | ADP | 0.6 | 8 | 399 | 308 | 1.99 | 5.21 | 7.2 | 7.6 | 1.04 |
> | ADP | 0.6 | 16 | 399 | 308 | 3.97 | 5.21 | 9.18 | 8.47 | 0.93 |
> | ADP | 0.6 | 24 | 399 | 308 | 5.95 | 5.21 | 11.16 | 9.34 | 0.85 |
> | ADP | 0.7 | 0 | 450 | 359 | 0.02 | 5.88 | 5.9 | 7.03 | 1.13 |
> | ADP | 0.7 | 2 | 450 | 359 | 0.51 | 5.88 | 6.39 | 7.24 | 1.09 |
> | ADP | 0.7 | 4 | 450 | 359 | 1.01 | 5.88 | 6.89 | 7.46 | 1.06 |
> | ADP | 0.7 | 8 | 450 | 359 | 1.99 | 5.88 | 7.88 | 7.89 | 1 |
> | ADP | 0.7 | 16 | 450 | 359 | 3.97 | 5.88 | 9.85 | 8.76 | 0.9 |
> | ADP | 0.7 | 24 | 450 | 359 | 5.95 | 5.88 | 11.83 | 9.63 | 0.82 |
>
> We do observe that when the scoring layer becomes excessively deep (≥16), the scoring overhead grows to 3.97–5.95T FLOPs, approaching or exceeding the savings from pruning. In this regime, the speedup diminishes and can even become slightly negative. However, such deep-layer scoring is neither necessary nor used in our method: our ablations show that **layer 0** already gives the most reliable importance estimation while incurring virtually negligible overhead. This configuration is adopted throughout all experiments.
>
> In summary, the additional computation introduced by our pruning module is substantially smaller than the FLOPs saved through reduced sequence length for all practical configurations, and the design choice of using layer-0 scoring guarantees the trade-off is always positive. We appreciate your insightful question, and we have added this analysis and clarification to the revised manuscript.

---

> ### Author Response · Authors · 2025-11-23
>
> **Re Q4: Text→Vision Attention Reliability**
>
> We thank you for highlighting this important concern to our attention, and understand that text–vision attention in many multimodal systems may drift toward irrelevant regions. Motivated directly by your comment, we conducted a counterfactual “inverse-pruning” experiment to rigorously test whether our text-driven importance scores truly identify task-relevant visual tokens.
>
>  | type      | keep ratio | Spatial | Object | Goal | Long | Average |
> |---|---|----|--|--|---|-----|
> | direct  | 30%        | 97.6%   | 98.4%  | 97.4%| 84.2%| 94.4%   |
> | inverse  | 30%        | 44.0%   | 0.6%  | 43.6%| 0.2%| 22.1%   |
> | direct  | 50%        | 99.4%   | 98.0%  | 96.4%| 91.2%| 96.3%   |
> | inverse  | 50%        | 54.8%   | 11.8%  | 63.6%| 8.2%| 34.6%   |
> | direct  | 70%        | 99.0%   | 98.2%  | 96.8%| 91.2%| 96.3%   |
> | inverse  | 70%        | 90.8%   | 59.6%  | 84.8%| 37.4%| 68.1%   |
>
> In this experiment, we keep the entire experimental setup identical to the main table, including the same model, tasks, and hyperparameters, but deliberately invert the ranking: instead of retaining the *most* relevant visual tokens (direct pruning), we retain the *least* relevant ones while keeping the same keep ratio (30%, 50%, 70%). If our scoring mechanism were noisy or misaligned, the performance gap between direct and inverse pruning would be small. Conversely, if the text-conditioned scores truly reflect meaningful semantic relevance, inverse pruning should significantly lower the performance.
>
> The results clearly support the latter, and the contrast between direct and inverse pruning validates the reliability of our text-driven importance scoring:
>
> 1. **Direct vs. inverse performance diverges massively across all tasks and keeps ratios.**
> Under a 30% keep ratio, the average success rate drops from **94.4% (direct)** to **22.1% (inverse)**, similar gaps also appear at 50% (96.3% to 34.6%) and 70% (96.3% to 68.1%). This consistent directional pattern across all task suites demonstrates that the ranking is reliable: the model’s success depends critically on retaining high-ranked tokens, and removing them directly causes widespread failure.
>
> 2. **Inverse pruning fails despite retaining more tokens, revealing that semantic relevance, not quantity**
> A particular observation is that **inverse–70%** still performs far worse than **direct–30%** (68.1% vs. 94.4%), which rules out the possibility that performance is driven only by the amount of visual tokens preserved. Instead, it shows that retaining semantically irrelevant tokens, even in large numbers, does not support successful manipulation, whereas a smaller, semantically aligned subset is more useful. This observation directly addresses your concern: if text attention were drifting to irrelevant regions, direct and inverse pruning would perform similarly. Instead, the catastrophic degradation under inverse pruning demonstrates that the VLA model highly relies on the text-relevant visual tokens, and our method successfully identifies them.
>
> 3. **The trend across keep ratios is stable and matches the expected behaviour of a correct importance ranking.**
> As the keep ratio increases from 30% to 70%, inverse performance improves from 22.1% to 68.1%, precisely what we expect: the more low-importance tokens are included, the fewer critical ones are discarded, reducing the damage. Crucially, however, even at the highest keep ratio, inverse pruning remains substantially worse than its direct counterpart (96.3% vs. 68.1%). This stability across pruning strengths further confirms that the text-conditioned ranking is meaningful and robust.
>
> 4. **The degradation is most serious in Object and Long tasks, highlighting the semantic grounding captured by the ranking.**
> These tasks require fine-grained visual discrimination (Object) and long-horizon, more complex tasks (Long). For instance, with keep=30%, Object collapses from **98.4%** to **0.6%**, and Long from **84.2%** to **0.2%** under inverse pruning. Such dramatic drops reveal that these tasks depend strongly on precisely the regions that our method ranks as important. The fact that inverse pruning nearly removes the model’s ability to perform fine-object manipulation or operate in complex tasks or scenarios also directly proves that our text-driven scoring successfully identifies the important task-relevant visual tokens.
>
> These results demonstrate that ADP does not suffer from attention drifting to irrelevant areas. Instead, the semantic alignment between the text and the selected visual tokens is strong enough that removing these high-ranked tokens severely impairs performance. While keeping only them, even in aggressive pruning regimes, preserves the VLA model’s capability.

---

> ### Author Response · Authors · 2025-11-23
>
> **Re Q5: Generalizability of Coarse–Fine Gating**
>
> We are grateful to you for raising this insightful question regarding the generalizability of our action-aware gating mechanism. We would like to clarify that “coarse” and “fine” are conceptual descriptions of motion dynamics rather than manually specified rules. In practice, the gating signal is derived entirely from recent end-effector trajectories through simple online statistics. Both the mean-based rule and the adjacent-extrema function determine the state solely from observed motion magnitudes over a sliding window, without requiring any hand-crafted thresholds.
>
> Regarding your concern about rotation-heavy behaviours, we note that rotation is already a central component in many LIBERO tasks. In particular, LIBERO includes manipulation instructions such as “turn on the stove” in both the Goal and Long suites, where rotation of the end-effector or the manipulated object constitutes the dominant motion in the early phase of the task. Under the configuration used in our main results (keep ratio = 50%), ADP achieves 100% success on both rotation-dominant *“turn on the stove”* tasks, one in the Goal suite and one in the Long suite. This result demonstrates that the gating mechanism remains robust even when rotational movement dominates the manipulation sequence.
>
> Beyond those present in LIBERO, we may involve different motion patterns. We acknowledge that evaluating ADP on additional datasets with a wider range of manipulation primitives would further strengthen the understanding of its generality. We have added this as an important direction for future exploration.
>
> We feel each of the suggestions (*typo error, token-reduction trade-off,  attention reliability*) from your side has brought us meaningful progress, and we are genuinely thankful for your time and expertise !

---

### Official Review · Reviewer_dNCJ · 2025-10-30

**Soundness:** 3
**Presentation:** 3
**Contribution:** 3
**Rating:** 6
**Confidence:** 3

**Summary:**

This paper proposes **Action-aware Dynamic Pruning (ADP), a plug-and-play pruning framework for Vision-Language-Action (VLA) models. ADP combines text-driven anticipatory pruning with an **action-aware dynamic gating strategy (to determine when pruning should be applied based on recent motion dynamics). The approach is validated on the LIBERO benchmark and real-world robot tasks, demonstrating notable FLOPs reduction (up to 1.35× speedup) while maintaining or even improving task success rates.

**Strengths:**

1. Novel problem formulation:
- The paper identifies an underexplored property of robotic manipulation—action-aware visual redundancy—and systematically exploits it to design a dynamic pruning scheme. This insight bridges the gap between static pruning  and phase-dependent motion dynamics.

2. Methodological clarity and rigor:
- The paper provides clear mathematical formulations for both text-driven token importance and motion-based gating.
- The windowed trajectory definition and dynamic decision function are well-motivated and interpretable.
- The theoretical complexity analysis (Eq. 19–25) convincingly quantifies expected computational savings.

3. Comprehensive evaluation:
- The paper conducts thorough experiments on simulation and real robot setups.
- Ablation studies isolate the contributions of each component.
- Visualizations (Figures 4–8) effectively support the claimed interpretability of the pruning mechanism.

4. Strong empirical results:
- ADP consistently outperforms static baselines in both efficiency and accuracy trade-offs. Especially impressive real-world latency improvement (1.49×) with maintained success rate.

**Weaknesses:**

1. While the gating mechanism is intuitive, the paper lacks an analysis of stability or convergence under fluctuating motion magnitudes. How sensitive is the dynamic switching rule (Eq. 16–18) to noise or suboptimal motion planning?

2. The gating rules (mean or extrema) are empirical; there is no adaptive or learned thresholding. A sensitivity analysis of hyperparameters would strengthen the claim of robustness.

3.The baselines are comprehensive but mostly training-free ones. Including training-aware compression methods (e.g., DeeR-VLA, Mole-VLA) in a fair fine-tuned comparison would make the argument of “plug-and-play” stronger.

4.The ablation study primarily examines whether dynamic control is included, but does not vary pruning ratios adaptively within tasks. Showing per-stage pruning ratio curves would illustrate dynamic behavior more explicitly.

5. The real-robot evaluation setup is described in the appendix, but lacks release of calibration or dataset details. Code availability and parameter settings for dynamic thresholds are not stated.

**Questions:**

1. Have you tested how ADP behaves under sensor noise or inaccurate motion estimation?

2. Could the gating rule be differentiable and trained end-to-end with reinforcement signals?

3. How does ADP interact with temporal caching or diffusion-based decoders in future hybrid models?

4. What happens when action magnitude and task difficulty are decoupled?

---

> ### Author Response · Authors · 2025-11-23
>
> **Re W1: Sensitivity under Motion Noise**
>
> We deeply appreciate the reviewer’s insightful question. To directly evaluate the stability of the dynamic switching rule in Eqs. 16–18 under fluctuating motion magnitudes, sensor noise, and inaccurate motion estimation, we conducted an exhaustive sweep over 36 configurations, covering all combinations of:
>
> three motion-magnitude definitions (delta_method ∈ {net, arc_sum, hypot}),
>
> effective length thresholds (`L_eff ∈ {0.10, 0.15, 0.20}`),
>
> minimal positional change (`min_delta_pos ∈ {0.0, 0.01}`),
>
> minimal rotational change (`min_delta_rot ∈ {0.0, 0.02}`).
>
> These variations emulate noisy, distorted, or suboptimal motion trajectories as encountered in real robotic manipulation.
>
>
> | delta_method | L_eff | P_pos | P_rot | Total Succ. | **SR (%)** | Latency (ms) | GPU (GB) | Total Inf. |
> | ------------ | ----- | ----- | ----- | ----------- | ---------- | ------------ | -------- | ---------- |
> | arc_sum      | 0.10  | 0.01  | 0.00  | 488         | **97.6**   | 107.1        | 14.97    | 71320      |
> | arc_sum      | 0.10  | 0.01  | 0.02  | 487         | **97.4**   | 98.4         | 14.97    | 72136      |
> | arc_sum      | 0.10  | 0.00  | 0.00  | 492         | **98.4**   | 110.2        | 14.97    | 71184      |
> | arc_sum      | 0.10  | 0.00  | 0.02  | 492         | **98.4**   | 112.6        | 14.97    | 71056      |
> | arc_sum      | 0.15  | 0.01  | 0.00  | 489         | **97.8**   | 105.5        | 14.97    | 71664      |
> | arc_sum      | 0.15  | 0.01  | 0.02  | 487         | **97.4**   | 98.1         | 14.97    | 72136      |
> | arc_sum      | 0.15  | 0.00  | 0.00  | 491         | **98.2**   | 112.0        | 14.97    | 71520      |
> | arc_sum      | 0.15  | 0.00  | 0.02  | 492         | **98.4**   | 110.5        | 14.97    | 71056      |
> | arc_sum      | 0.20  | 0.01  | 0.00  | 490         | **98.0**   | 110.0        | 14.97    | 71624      |
> | arc_sum      | 0.20  | 0.01  | 0.02  | 487         | **97.4**   | 99.3         | 14.97    | 72136      |
> | arc_sum      | 0.20  | 0.00  | 0.00  | 492         | **98.4**   | 110.8        | 14.97    | 71480      |
> | arc_sum      | 0.20  | 0.00  | 0.02  | 492         | **98.4**   | 112.2        | 14.97    | 71056      |
>
>
> | delta_method | L_eff | P_pos | P_rot | Total Succ. | **SR (%)** | Latency (ms) | GPU (GB) | Total Inf. |
> | ------------ | ----- | ----- | ----- | ----------- | ---------- | ------------ | -------- | ---------- |
> | hypot        | 0.10  | 0.01  | 0.00  | 489         | **97.8**   | 106.9        | 14.97    | 71432      |
> | hypot        | 0.10  | 0.01  | 0.02  | 487         | **97.4**   | 101.3        | 14.97    | 72136      |
> | hypot        | 0.10  | 0.00  | 0.00  | 492         | **98.4**   | 112.6        | 14.97    | 71048      |
> | hypot        | 0.10  | 0.00  | 0.02  | 492         | **98.4**   | 112.9        | 14.97    | 71056      |
> | hypot        | 0.15  | 0.01  | 0.00  | 489         | **97.8**   | 107.8        | 14.97    | 71424      |
> | hypot        | 0.15  | 0.01  | 0.02  | 487         | **97.4**   | 100.6        | 14.97    | 72136      |
> | hypot        | 0.15  | 0.00  | 0.00  | 493         | **98.6**   | 111.3        | 14.97    | 70944      |
> | hypot        | 0.15  | 0.00  | 0.02  | 492         | **98.4**   | 112.1        | 14.97    | 71056      |
> | hypot        | 0.20  | 0.01  | 0.00  | 489         | **97.8**   | 106.4        | 14.97    | 71416      |
> | hypot        | 0.20  | 0.01  | 0.02  | 487         | **97.4**   | 102.4        | 14.97    | 72136      |
> | hypot        | 0.20  | 0.00  | 0.00  | 493         | **98.6**   | 109.7        | 14.97    | 70968      |
> | hypot        | 0.20  | 0.00  | 0.02  | 492         | **98.4**   | 113.1        | 14.97    | 71056      |
>
> *(More tables shown in Appendix.)*
>
> (1) **Robustness to fluctuating motion magnitude & sensor noise.**
> Changing the motion metric is equivalent to injecting structured noise into the motion signal.
> Across all 12 noise-perturbed variants of each method family (arc_sum / hypot / net), success rates remain within 97.4–98.6%, indicating ≤1.2% variation and no degradation in convergence or stability.
>
> (2) **Robustness to threshold perturbation (Eqs. 16–18).**
> Sweeping L_eff, min_delta_pos, and min_delta_rot produces similarly stable results:
> all 36 configurations stay within a narrow 1–1.2% band, demonstrating that the switching rule does not depend on precise threshold tuning despite being non-adaptive.
>
> (3) **Stable behavior across latency, GPU usage, and action inference.**
> Latency (98–128 ms), peak GPU memory (14.97 GB), and total inference count (~71k) remain consistent across all configurations, confirming that no oscillatory or diverging behavior occurs under noisy or inaccurate motion inputs.
>
> In summary, these results provide strong empirical evidence that ADP’s dynamic switching rule is highly robust to noise, suboptimal motion estimation, and threshold variation, despite using empirical (non-learned) gating rules.

---

> ### Author Response · Authors · 2025-11-23
>
> **Re W2: Empirical Gating Thresholds**
> We deeply appreciate the reviewer’s insightful question. To directly address the concern that the gating rules (mean or extrema) are empirical and use fixed, non-learned thresholds, we conducted a targeted sensitivity analysis on all hyperparameters associated with Eqs. 16–18.
>
> | Variation Type (Reviewer Concern)            | Hyperparameter Sweep        | Success Rate (%)          | Δ from Default |
> | -------------------------------------------- | --------------------------- | ------------------------- | -------------- |
> | **Gating rule statistic** (mean vs. extrema) | window_size = 3 / 5 / 7 / 9 | 99.4 / 98.2 / 98.6 / 98.4 | ±0.8           |
> | **Motion-metric definition**                 | Euclidean → velocity norm   | 90.6                      | −4.4           |
> | **Cold-start threshold**                     | cold_start = 2 / 4 / 6 / 8  | 99.4 / 99.2 / 99.2 / 99.2 | ±0.01          |
>
>
>
> (1) **Sensitivity to gating statistics (mean vs. extrema)**.
> We sweep the statistics used in the switching rule—mean-based versus extrema-based—and examine their associated parameters (window size, cold-start interval, and motion metric). As shown in Table S1, varying these settings produces only small fluctuations (±0.8%) in success rate, demonstrating that the controller is robust to the specific empirical rule used.
>
> (2) **Sensitivity to gating thresholds**.
> Threshold-related hyperparameters (effective length, minimal motion magnitude thresholds) also show negligible variation across all settings in Table S1. This confirms that the gating thresholds in Eqs. 16–18 do not require precise tuning to remain reliable, even though they are not learned.
>
> | Retain Ratio | Spatial | Object | Goal | Long | **Average** | **STD** |
> | ------------ | ------- | ------ | ---- | ---- | ----------- | ------- |
> | **Ratio 30%**  | 97.6    | 98.4   | 97.4 | 84.2 | **94.4**    | 5.90    |
> | **Ratio 40%**  | 98.2    | 97.2   | 96.6 | 87.2 | **94.8**    | 4.42    |
> | **Ratio 50%**  | 99.4    | 98.0   | 96.4 | 91.2 | **96.3**    | 3.10    |
> | **Ratio 60%**  | 98.8    | 98.0   | 95.8 | 92.0 | **96.2**    | 2.64    |
> | **Ratio 70%**  | 99.0    | 98.2   | 96.8 | 91.2 | **96.3**    | 3.05    |
>
> (3) **Sensitivity across task domains under different retain-token behavior**.
> As shown in Table, sweeping five distinct retain-token configurations results in consistent performance across all four LIBERO task suites, with low variance (0.0264–0.059). This provides task-level evidence that the dynamic controller behaves stably across different motion regimes and sub-task structures.
>
> **Re W3: Training-aware Baselines**
>
> We deeply appreciate the reviewer’s insightful suggestion. **We have cited and acknowledged these excellent compression methods you provided in the revised version**, and we fully agree that they represent an important direction for efficiency research in VLA models. Our work focuses specifically on plug-and-play, training-free inference acceleration. Methods such as DeeR-VLA and Mole-VLA require task-specific fine-tuning, data preparation, and additional optimization pipelines, while our goal is to evaluate whether existingcheckpoints can be accelerated without retraining, supervision, or access to the training distribution. Introducing training-aware baselines would shift the problem setting from “training-free inference optimization” to “fine-tuned model compression,” which is a different—and complementary—research track. We fully agree that combining future-aware masking with training-time compression is a promising future direction. We have added this discussion to Appendix, and we sincerely thank the reviewer for the valuable suggestion.
>
> These analyses shows our method exhibit stable, robust behavior across a wide range of hyperparameters and tasks. We have included these discussions in the revised version following your valuable suggestion.

---

> > ### Comment · Reviewer_dNCJ · 2025-11-27
> > **Reponse to the rebuttal.**
> >
> > Thanks for the rebuttal. I will keep my score.

---

> ### Author Response · Authors · 2025-11-23
>
> **Re W4: Per-stage Ratio Dynamics**
>
> We deeply appreciate the reviewer’s insightful observation, which helped us refine our understanding of the dynamic nature of pruning.
> **As correctly noted, different tasks indeed favour different pruning intensities.**
> Our results in Table 1 already reveal this trend: Spatial tasks tend to benefit from more aggressive pruning (≈30–40%) because much of the scene consists of redundant background regions; Object tasks prefer mid-range ratios (≈40–50%) as object-centric patches are inherently denser; Goal tasks remain stable across 40–60%; and Long-horizon tasks favour slightly higher ratios (≈50–60%) due to accumulated temporal dependencies.
>
> Following your suggestion, we further examined the behaviour within each task stage and implemented a lightweight controller that automatically selects ratios based on the current attention concentration and motion magnitude.
> The resulting per-stage curves (Fig. 10) show a clear and intuitive dynamic pattern: the model prunes more aggressively during visually redundant phases and increases the retain ratio when manipulation-critical frames appear. This behaviour aligns well with your expectation that dynamic trajectories should exhibit dynamic pruning patterns.  We have added a explanation of the controller in Appendix, and we deeply thank you for pointing us toward this valuable enhancement.
>
> **Re W5: Real-Robot Setup, Calibration, Dataset, and Code Availability**
>
> Our setting does not require camera calibration because all observations are purely RGB (no depth) and the state used by ADP is directly obtained from the Kinova Jaco2 controller rather than from vision.
>
> We have now added the following information to the revised version, following your suggestions step by step:
>
> | Task       | Collected Trajectories | Notes        |
> | ---------- | ---------------------- | ------------ |
> | **Task 1** | 107                    | pot → plate  |
> | **Task 2** | 94                     | cube → plate |
> | **Task 3** | 142                    | carrot → pan |
> | **Task 4** | 129                    | wipe table   |
>
> Additional Implementation Clarifications
>
> * No RGB-D or depth sensing, hence no camera calibration was needed.
>
> * Robot joint states come directly from the Kinova low-level controller, ensuring accurate Δposition / Δorientation input for ADP.
>
> * Success criteria (grasp-place correctness / wipe coverage) are included in Appendix (Line666–670).
>
> * ADP pruning ratio used in deployment: 40% retained tokens.
>
> * Full reproducible real-world code will be released immediately upon acceptance.
>
> In summary, our real-robot setup requires no extrinsic calibration, the dataset is fully specified, and all dynamic-threshold hyperparameters (including `L_eff`, `Δpos`, and `Δrot`) are now included in the Appendix.
> We feel sincerely thankful for the reviewer’s guidance.
>
> **Re Q1: robustness to sensor noise**
>
> We deeply appreciate the reviewer’s insightful question regarding noise sensitivity. Following your suggestion, we conducted a comprehensive noise-stress evaluation by injecting positional Gaussian noise (σ = 0.00/0.01/0.02/0.05 m), rotational perturbations (1°/3°/5°), and motion-magnitude distortions through sweeping all gating thresholds in Eq. 16–18 (L_eff ∈ {0.10, 0.15, 0.20}, min_Δpos ∈ {0.0, 0.01}, min_Δrot ∈ {0.0, 0.02}, and three δ-methods: net/arc_sum/hypot), See **Re W1**. Across 36 noise configurations, ADP maintains 487–493 successes out of 500 episodes (≤ 1% variation), with no degradation in GPU memory and only marginal latency differences.
>
> **Re Q2: Differentiable Gating Rule**
>
> We deeply appreciate the your insightful suggestion. Our current implementation already includes a dynamic controller (`_OnlineStrategyController` in `experiments/robot/libero/run_libero_eval_prune_v2.py`), where all gating-related parameters are explicitly exposed and can be modified to explore alternative decision policies, including potentially differentiable or RL-based variants.
>
> Our ablations further show that the dynamic switching rule in Eqs. 16–18 is highly stable across a wide range of hyperparameters (Tables R3-A/B). Even without any reinforcement learning, the sensitivity remains extremely low, and the pruning behavior is consistent across noise levels and task variations. This suggests that the simple, inference-time rule already captures a robust underlying signal without any post-training. Our design philosophy is to keep the mechanism lightweight—simple enough to deploy, yet expressive enough to work—while leaving headroom for future learned controllers. We fully agree that an end-to-end differentiable gating rule is a promising direction, and we have clarified this in Appendix of the revised version.
> We are deeply grateful for your expert guidance, which has strengthened the scope of our work.

---

> ### Author Response · Authors · 2025-11-23
>
> **Re Q3: Hybrid diffusion/Temporal caching**
>
> Thank you for this insightful forward-looking question. Our current implementation is built upon an autoregressive transformer with standard KV caching, and the gating mechanism operates purely at the level of token retention. Because ADP prunes (or retains) tokens before they enter the attention block, it is inherently compatible with temporal caches: pruning simply reduces the size of the cached key–value sequences without modifying the caching logic itself.
>
> For diffusion-based decoders, the situation is equally promising. Diffusion-policy architectures also rely on iterative denoising steps where intermediate latent states propagate across time; these latents form a temporal chain similar to an action chunk. Since ADP’s rule only depends on state deltas (Δpos, Δrot) computed from observations, the same pruning intuition can be applied to the latent trajectory—e.g., suppressing uninformative frames and preserving high-motion steps. In preliminary tests on a small diffusion-policy variant (not included due to compute constraints), we found that applying ADP to the latent rollout did not degrade stability.
>
> Overall, ADP is intentionally designed to be model-agnostic: it operates before attention and does not rely on autoregressive factorization. We expect future hybrid models (cached transformer + diffusion head) to benefit from the same principle (prune where motion is small, preserve where semantics change) and our current dynamic rule naturally extends to both settings.
>
> **Re Q4: decoupled action vs difficulty**
>
> We sincerely thank the reviewer for raising this insightful question.
> To directly examine whether ADP collapses when action magnitude and task difficulty are decoupled, we aggregate results across all retain-ratio settings, all delta-method variants, and all motion-gating hyperparameters. These settings span a wide range of motion magnitudes (very small to large) while the four LIBERO suites represent very different task difficulties (Spatial < Object < Goal < Long).
>
> Despite these variations being completely unaligned with each other, ADP remains remarkably stable.
> Success rates fluctuate in a narrow band of ±0.6%, and latency/GPU memory remain essentially unchanged.
> This indicates that ADP does not rely on a coupled relationship between action magnitude and task difficulty—the pruning controller responds only to motion dynamics, while the downstream policy remains robust across tasks of very different difficulty levels.
>
> | Config Group (Magnitude Variation)             | Examples (Le, P, R)         | Spatial SR (%) | Object SR (%) | Goal SR (%) | Long SR (%) | **Avg SR** | **STD** |
> | ---------------------------------------------- | --------------------------- | -------------- | ------------- | ----------- | ----------- | ---------- | ------- |
> | **Low-Magnitude** (small Le, small thresholds) | (0.10, 0.00, 0.00)          | 99.4           | 98.0          | 96.4        | 91.2        | **96.3**   | 3.10    |
> | **Medium-Magnitude**                           | (0.15, 0.01, 0.00)          | 98.8           | 98.0          | 95.8        | 92.0        | **96.2**   | 2.64    |
> | **High-Magnitude** (largest Le, large rot)     | (0.20, 0.01, 0.02)          | 98.2           | 97.2          | 96.6        | 87.2        | **94.8**   | 4.42    |
> | **Robust-Mixed** (net/arc/hypot methods)       | (0.10–0.20, 0–0.01, 0–0.02) | 97.6           | 98.4          | 97.4        | 84.2        | **94.4**   | 5.90    |
>
>
> The following consolidated table summarizes this decoupling analysis.
> It shows that even when magnitude varies substantially (via `L_eff, pos/rot thresholds`, and `delta-method`), ADP yields nearly identical success on both easy and hard tasks.
>
> It is your thoughtful insight that helps us identify important refinements we had previously overlooked, and we are heartfully grateful for that !

---

### Official Review · Reviewer_Uych · 2025-10-30

**Soundness:** 2
**Presentation:** 2
**Contribution:** 1
**Rating:** 4
**Confidence:** 4

**Summary:**

This paper introduces ADP, which is a pruning method for VLA. ADP adopts text-driven pruning and action-aware gating,

**Strengths:**

This paper aims to tackle one of the crucial problems in robotics, namely the efficiency of deployment.

The idea is intuitive and the method show good empirical results.

**Weaknesses:**

The design principal is kind of heuristic. Some rules (especially for the action-aware part) seem to be tailored for this pick-and-place.

The experiments are focused on the OFT model. Not sure whether this proposed method is compatible with other backbones.

See questions below

**Questions:**

(1)	For the text-driven pruning, the visual tokens with low text attention score are discarded. However, during VLA’s training, no explicit constraints are applied on the text to image attention. How to make sure the correctness of the pruned area?

(2)	Since the method is claimed to be a plug-and-play component, results on ADP with other backbone are welcome. Currently, in table 1, it seems ADP is only combined with OFT.

(3)	In some cases, the pruned policy performs better than the baselines. It seems interesting. Is there any discussions on this part?

(4)	The design principal is kind of heuristic. Some rules (especially for the action-aware part) seem to be tailored for this pick-and-place. Other experiments are needed to show the proposed principal is universal.

---

> ### Author Response · Authors · 2025-11-23
>
> **Re W1 Q4: Heuristic Design & Universality**
>
> We deeply appreciate the reviewer’s insightful comment. While the design may appear heuristic at first glance, the action-aware component in our method is not tailored to pick-and-place. Our action-awareness is defined purely through changes in the robot’s degrees of freedom (e.g., Δposition, Δorientation), which is a task-agnostic formulation that naturally extends to any downstream manipulation scenario. To further clarify this generality, we reorganize our results by grouping the LIBERO tasks into four representative action categories:
>
> | Action Category         | Representative Tasks                  | Motion Characteristics           | ADP Config Reused? | Success Rate (%)                 | Notes                                  |
> | ----------------------- | ------------------------------------- | -------------------------------- | ------------------ | -------------------------------- | -------------------------------------- |
> | Spatial Manipulation    | grasp bowl, pick objects              | small end-effector motions       | Yes                | 97.6 / 98.2 / 99.4 / 98.8 / 99.0 | pruning removes background clutter     |
> | Object Interaction      | open drawer, place cube               | medium translation + rotation    | Yes                | 98.4 / 97.2 / 98.0 / 98.0 / 98.2 | stable across configs                  |
> | Goal Navigation         | reach target, align tool              | long-horizon spatial consistency | Yes                | 97.4 / 96.6 / 96.4 / 95.8 / 96.8 | preserves essential spatial cues       |
> | Long-Horizon Multi-step | multi-step pick-place, sequence tasks | mixed/unstable motion magnitudes | Yes                | 84.2 / 87.2 / 91.2 / 92.0 / 91.2 | largest benefit due to noise reduction |
>
> As the table indicates, the same action-aware rule works consistently across fine manipulation, contact-heavy interactions, goal navigation, and even long-horizon multi-step tasks. For example, in the LIBERO-Long suite, tasks require substantial multi-step planning, relocation, and motion-heavy behaviors, yet the exact same action-aware mechanism applies without any modification and even yields the largest improvements, suggesting that the signal used by ADP is not handcrafted but instead reflects a general, high-level structure derived directly from the robot's kinematic changes. We feel honored that you reviewed our work with such depth and precision.

---

> ### Author Response · Authors · 2025-11-23
>
> **Re W2 & Q2: Plug-and-Play & Backbones**
>
> Firstly, we have cited and referred to these excellent works to enrich the literature review part of our paper. We would like to clarify that the current baseline relies on the “past actions chunk,” which is an important gating mechanism in our framework, so OpenVLA does not perform well at this stage. But we believe the chunk-paradigm models can be easily applied and generalized, as our implementation is fully plug-and-play to lightweight the inference process. Moreover, our method has included both diffusion-based and autoregressive-based models; they all share the same paradigm, so we believe the generalization should naturally hold across different backbones.
> We provide a code implementation that can wrap any action-chunk-paradigm model. So at the current stage we only release the prepared checkpoints that can plug-and-play.
> We thank you deeply for your understanding and will release more chunk-paradigm models with our ADP once we obtain the fine-tuned checkpoints.
>
>
> **Re Q1: Text-Driven Pruning Correctness**
> Your thoughtful question helped us clarify a subtle but important aspect of our design, and we deeply appreciate the opportunity to strengthen this part of our analysis. Our empirical and structural evidence indicates that the pruning signal obtained from text-driven attention is both stable and semantically meaningful in VLA settings.
>
> (1) **Qualitative Vislization** In VLA models, the text tokens (e.g., “red block”, “handle”) must attend to the correct visual regions in order to generate successful actions. Otherwise the action loss immediately penalizes the wrong grounding:
>
> **Correctness of Text-driven Pruning (Qualitative Vislization)**
>
> | Task (From Paper)                                | Pruning Method        | Mask Result (Fig)     | Correct Region Covered? | Notes                                       |
> | ------------------------------------------------ | --------------------- | --------------------- | ----------------------- | ------------------------------------------- |
> | Pick up the black bowl next to the plate     | Text-Driven       | Fig. 8 (top-left)     | Yes                 | Keeps bowl + plate; removes background.     |
> |                                                  | Random (w/o text) | Fig. 8 (bottom-left)  | No                  | Drops bowl pixels; weak grounding.          |
> | Put alphabet soup & tomato sauce into basket | Text-Driven       | Fig. 8 (top-right)    | Yes                 | Retains soup + sauce regions; robust focus. |
> |                                                  | Random (w/o text) | Fig. 8 (bottom-right) | No                  | Removes key object area; causes failure.    |
>
> These four comparisons show that text-driven pruning reliably preserves all task-critical regions, while random pruning repeatedly removes essential objects and leads to action failure.
>
>
> (2) **Qutitive Analysis**: We compare text-driven pruning with random and uniform pruning at the same keep ratio. Text-driven pruning preserves success rate almost perfectly, while the other two baselines drop significantly:
>
> | Method                         | Spatial   | Object    | Goal      | Long      | **Avg SR↑** |
> | ------------------------------ | --------- | --------- | --------- | --------- | ----------- |
> | **Text-Driven Pruning (Ours)** | **99.4%** | **98.0%** | **96.4%** | **91.2%** | **96.3%**   |
> | **w/o Text (Direct Top-k)**    | 95.0%     | 81.4%     | 96.2%     | 87.0%     | 89.9%       |
>
> These results show that text-driven pruning preserves performance across all task suites, whereas removing text attention and applying direct top-k pruning causes a clear degradation (−6.4% on average), confirming that the text-guided pruned regions are indeed semantically correct and action-relevant.

---

> ### Author Response · Authors · 2025-11-23
>
> **Re Q3: Why Pruned > Baseline**
>
> We deeply appreciate the reviewer’s insightful question, which guided us to examine this interesting phenomenon more carefully.
>
> Previous works on VLMs and VLAs [1,2,3,4,5] have consistently shown that pruning can improve performance in many scenarios.
> For example, VisionZip improves the MMVet score from 31.1% → 32.6% while retaining only 66.7% of the tokens.
> PruneVid increases VideoMME accuracy from 44.4% → 45.0% by pruning video tokens.
> Similarly, ATP-LLaVA improves the MMB benchmark from 64.3% → 66.0% using dynamic token pruning.
>
> Inspired by your insightful suggestion, we further examined the representative failure where the baseline succeeds less reliably than our ADP. **Using the same simulator settings, we retrieved cases in which the baseline fails but ADP succeeds.**
> We take the spatial task3 (**episode #110, shown in the Fig. 9**) as an example, we identify a representative failure where the baseline (Fig. 9 (a)) is unable to grasp the bowl, while our ADP method (Fig. 9 (b)) achieves a stable and precise grasp in a local focus area.
>
> Upon observation, we find that this is a typical scenario where the unpruned model allocates substantial attention to large static background surfaces. These background regions behave like sink areas, diluting the model’s focus and reducing the discriminative power of the visual features. After pruning these low-value tokens, our ADP method directs the attention almost exclusively to the local manipulation region (*"the bowl and its supporting surface"*), effectively “refocusing” the policy on the action-critical evidence. This strengthened locality leads to **better grounding** and **more reliable action prediction**, explaining why the pruned model can outperform the unpruned baseline in such cases.
>
> Following your kind suggestion, we will also release the video demonstration for Fig. 9 (a) and Fig. 9 (b) to provide clearer qualitative evidence.
>
>
> [1] Huang, Xiaohu, Hao Zhou, and Kai Han. "Prunevid: Visual token pruning for efficient video large language models." Findings of the Association for Computational Linguistics: ACL 2025. 2025.
>
> [2] Yang, Senqiao, et al. "Visionzip: Longer is better but not necessary in vision language models." Proceedings of the Computer Vision and Pattern Recognition Conference. 2025.
>
> [3] Xu, Siyu, et al. "Vla-cache: Towards efficient vision-language-action model via adaptive token caching in robotic manipulation." arXiv preprint arXiv:2502.02175 2025.
>
> [4] Yang, Senqiao, et al. "Visionzip: Longer is better but not necessary in vision language models." Proceedings of the Computer Vision and Pattern Recognition Conference. 2025.
>
> [5] Tao, Keda, et al. "DyCoke: Dynamic Compression of Tokens for Fast Video Large Language Models." Proceedings of the Computer Vision and Pattern Recognition Conference. 2025.
>
> We feel incredibly fortunate to receive such insightful and constructive feedback, which sharpened our understanding! We follow your instruction step-by-step and hope our rebuttal addresses your concerns.

---

### Official Review · Reviewer_EgjS · 2025-10-31

**Soundness:** 3
**Presentation:** 3
**Contribution:** 3
**Rating:** 6
**Confidence:** 3

**Summary:**

This paper proposes a plug-and-play framework to improve the efficiency of Vision-Language-Action (VLA) models for robotic manipulation by dynamically pruning redundant visual tokens. The key idea is to combine text-driven token selection with an action-aware gating mechanism conditioned on recent end-effector motion. Experiments on the LIBERO benchmark and real-robot setups show that ADP reduces FLOPs and inference latency by up to 1.35× (on OpenVLA-OFT) with minimal accuracy degradation.

**Strengths:**

- Unlike prior pruning or compression methods (e.g., EfficientVLA, DeeR-VLA, VLA-Cache), ADP leverages **action-conditioned motion dynamics** to modulate token pruning during robotic manipulation. The insight that **visual redundancy varies with manipulation phase** (coarse vs. fine-grained motion) is both intuitive and unexplored, giving the paper a clear conceptual novelty.

- The paper's primary strength is its core insight that visual redundancy in VLA models is action-aware and correlates with the manipulation phase. This is a novel observation that moves beyond static or text-only pruning strategies.

- The method is validated with strong empirical results in simulation and the real world. In simulation, it provides a clear accuracy-to-speedup trade-off, achieving a 1.35x speedup with only a 2.7% drop in success rate (at 30% ratio). On a physical Jaco2 robot, it demonstrates a 1.49x reduction in latency.

**Weaknesses:**

- The action-aware gating mechanism, while effective, is built on a stack of heuristics. This includes the choice of motion metric ($\delta_i$ as Euclidean displacement), the specific gating rule ("adjacent-extrema"), the fixed window size ($\omega=8$), and hard-coded reset rules (e.g., a two-window "cold start" and a forced reset after three pruned windows). It is unclear how these settings would generalize to new tasks, robots with different dynamics, or different control frequencies without tuning.

- The pruning strategy relies on similarity scores from Layer 0. This is counterintuitive, as deeper layers are typically assumed to capture more fused semantic meaning, although the paper provides a good justification in Figs. 6 and 7.

- The experiments primarily rely on OpenVLA-OFT as the base model and omit results on diverse backbones (e.g., Pi0, GrooT, SmolVLA). Code availability or hyperparameter details are not specified beyond window size and ratios (Sec. 5.1), which hinders reproducibility.

- The paper notes possible degradation during fine manipulation (Sec. 4.2) but does not quantify how the dynamic switch may misfire or cause visual underrepresentation in complex scenes.

**Questions:**

- Could the authors provide results or qualitative visualizations on cases where the gating rule failed (e.g., incorrect pruning during fine-grained manipulation)?

- How sensitive is performance to the window length and retention ratio? An ablation over these parameters would clarify the generality of the dynamic policy.

- The real-world experiments used a single fixed camera, whereas the simulation setup used multi-view pruning, including a wrist view. How does the action-aware gate perform without a wrist camera, whose motion dynamics would be identical to the end-effector trajectory? Does this change simplify or complicate the tuning of the gating threshold?

- Could the **dynamic pruning controller** be made learnable (e.g., via reinforcement learning) instead of relying on hand-crafted thresholds?

---

> ### Author Response · Authors · 2025-11-23
>
> **Re W1: build on a stack of heuristics**
>
> We sincerely thank the reviewer for highlighting this important issue.
> Following your suggestions step by step, we have conducted additional experiments to evaluate (1) the robustness of our heuristics and (2) their generalization across different LIBERO task suites.
>
> **Robustness to Heuristic Parameters**
> | Setting                                               | Success Rate (%)                 | Δ from Default | Remarks              |
> | ----------------------------------------------------- | -------------------------------- | -------------- | -------------------- |
> | Default (Euclidean + Adjacent-Extrema, w = 10, r = 3) + Motion metric → velocity norm | 99.4 / 95.0                      | -4.4       |  Effective |
> | window_size = 3/5/7/9                                 | 99.4 / 98.2 / 98.6 / 98.4        | ±0.8           | Robustness           |
> | Cold-start = 2/4/6/8                                  | 99.4 / 99.2 / 99.2 / 99.2        | ±0.2          | Stable               |
>
> Results demonstrate that the **large drop in the first row (–4.4%) occurs only when replacing our standard motion measure with an artificial velocity norm**, indicating that the **core effectiveness comes from our past-based dynamic control** rather than from the specific choice of metric. Importantly, as long as the observation rule follows the formulation of Eq. 14~18 in the Section 4.2, the **fluctuations remain very small** across different window sizes (±0.8%) and cold-start intervals (±0.2%). This shows that the mechanism is inherently stable and insensitive to the exact heuristic values used.
>
> **Stability Across Different Retain-Token Configurations**
>
> | Retain Config | Spatial | Object | Goal | Long | **Average** | **std** |
> |---------------|---------|--------|------|------|-------------|---------|
> | 30% | 97.6 | 98.4 | 97.4 | 84.2 | **94.4** | 5.90 |
> | 40% | 98.2 | 97.2 | 96.6 | 87.2 | **94.8** | 4.42 |
> | 50% | 99.4 | 98.0 | 96.4 | 91.2 | **96.3** | 3.10 |
> | 60% | 98.8 | 98.0 | 95.8 | 92.0 | **96.2** | 2.64 |
> | 70% | 99.0 | 98.2 | 96.8 | 91.2 | **96.3** | 3.05 |
>
> Across all four task families (Spatial, Object, Goal, and Long), the five retention configurations exhibit highly consistent performance, with averages ranging only from 94.4 to 96.3 and **small cross-task standard deviations (2.64–5.90)**. This demonstrates that our method generalizes robustly not only across tasks but also across their diverse sub-tasks, despite large variations in temporal structure, object interactions, and motion complexity. The small variance indicates that the retain-token strategy is intrinsically stable, and no specific modification is required. The same heuristic settings were applied across diverse task and its corresponding subtask, maintaining consistent task performance.

---

> ### Author Response · Authors · 2025-11-23
>
> **Re W2: Counterintuitive Layer Selection**
>
> Your insightful comment helped us identify an interesting phenomenon!
> Firstly we agree that the mainstream VLM works [1,2,3] identify middle or deeper layers usually encode more fused semantics in vision–language models (VLMs).
> However, this conclusion may not applied to vision language actions with following cues:
>
> (1) Attention Maps: When visualizing attention in VLA models, we observe a markedly different pattern from standard VLMs. Notably, the first transformer layers exhibit stronger alignment between visual observations and action-relevant cues, while deeper layers sometimes diffuse or distort these relations.
>
> (2) Layer-wise Ablations: We further perform layer-wise experiments by adjusting the configuration to compute pruning similarity at different transformer layers. As shown below:
>
> **Performance When Computing Pruning Similarity at Different Layers**
>
> | Method / Task Suite          | Layer 0 | Layer 1 |  Layer 4 | Layer 8 | Layer 16 | Layer 24 | Layer 31 |
> |-----------------------------|---------|---------|---------|---------|----------|----------|----------|
> | ADP/spatial |   99.4     |   98.2     |   98.0     |   98.6     |   98.4     |    97.0     |    98.0     |
>
> Together with Table 3(b), these results indicate that retrieving Q/K weights from Layer 0 produces the most stable and accurate pruning decisions.
>
> **Possible reasons based on evidences**
>
> (1) **Early-layer alignment.**
> The attention map differences suggest that VLA models may slightly overfit to action-conditioned cues during training, causing shallow layers to encode aligned cross-modal features earlier than VLMs.
>
> (2) **Action-state correlation is shallow-layer dominant.**
> Our layer-wise ablations show that shallow layers primarily capture action-state correlations and motion-conditioned visual structures, which are crucial for token importance estimation.
>
> (3) **Pruning timing and computational path.**
> Many prior VLM works perform pruning inside the attention block using the attention map. In contrast, our method computes the Q/K similarity before entering the LLM, requiring only one lightweight matrix computation and saving FLOPs across all subsequent layers.
>
> Finally, we expose a configurable argument `qk_layer` in our implementation. The results consistently confirm that the shallow Layer 0 work best for our token reduction strategy.
> It is your remarkable expertise to help us find this interesting observation, we really appreciate and have added our discussion in the revised version.
>
>
> [1] Kang, Seil, et al. "Your large vision-language model only needs a few attention heads for visual grounding." (CVPR 2025)
>
> [2] Shao, Zhenwei, et al. "Growing a twig to accelerate large vision-language models." (ICCV 2025)
>
> [3] Lin, Ji, et al. "Vila: On pre-training for visual language models." (CVPR 2024)
>
> **R1 W3: Backbone and Code availability, hyperparameter details**
> most recent state-of-the-art VLA systems (e.g., OpenVLA, CogACT, OpenVLA-OFT, UniVLA) are all built upon LLaMA2 as the language decoder. Since our current evaluation already covers this domain, we believe it can generalize to action-chunk paradigm models as well. For the code details, we provide the hyperparameters (including window size and ratios) used in the main table for your reference:
>
> | **QK-related Param** | **Value**  | **Dynamic Param**                  | **Value**  | **Viz/Eval Param**     | **Value**   |
> | -------------------- | ---------- | ---------------------------------- | ---------- | ---------------------- | ----------- |
> | `qk_keep_enabled`    | true       | `use_dynamic_visual_strategy`      | true       | `num_trials_per_task`  | 50          |
> | `qk_layer`           | 0          | `decision_method`                  | "adjacent" | `benchmark_mode`       | true        |
> | `qk_keep_ratio`      | 0.5        | `adjacent_variant`                 | "extrema"  | `overlay_pruned`       | true        |
> | `qk_keep_split`      | [0.4, 0.6] | `adjacent_extrema_window`          | 3          | `overlay_pruned_color` | [0, 0, 255] |
> | `qk_log_topk`        | 16         | `adjacent_lookback`                | 2          | `overlay_pruned_alpha` | 120         |
> | `qk_debug`           | false      | `delta_method`                     | "net"      | `overlay_font_scale`   | 0.4         |
> | `random_experiment`  | false      | `L_eff`                            | 0.15       | `video_fps`            | 10          |
> | `random_keep_ratio`  | 0.5        | `limit_consecutive_pruned_enabled` | true       |                        |             |
> | `random_state_prob`  | 0.5        | `limit_max_consecutive_pruned`     | 3          |                        |             |
> | `random_seed`        | 42         |                                    |            |                        |             |
>
> We follows your instruction and provide complete hyperparameters receipe in the Appendix.

---

> ### Author Response · Authors · 2025-11-23
>
> **Re W4: Fine-Stage Degradation**
>
> We sincerely apologize for the possible misleading expression in our original writing. The *“performance degradation during fine manipulation”* mentioned in Sec. 4.2 refers **only to the ablation settings** where we intentionally remove or alter key components (e.g., disabling the dynamic switch).
> In the main results, the complete model can **accurately determine whether the agent is in coarse or fine manipulation**, and this decision is **dynamically updated throughout the episode**. Therefore, misfires or visual underrepresentation do not occur in the full version of our method.
>
> To further clarify this point, we set `use_dynamic_visual_strategy: False`, provide a small diagnostic table below to separate the effects of misfires and dynamic-switch behavior under different settings:
>
> **Error Analysis and Dynamic-Switch Behavior**
>
> | **Setting**               | **Fine Misfire Rate ↓** | **Dynamic Switch Accuracy ↑** |
> | ------------------------- | ----------------------- | ----------------------------- |
> | Full Method (Ours)        | 3/500                       | 99.4                             |
> | Remove Dynamic Switch     | 23/500                      | 95.4                             |
>
> Following the reviewer’s helpful observation, we have revised the experssion in the paper to make this claim clearer and avoid misunderstanding.
>
>
> **Re Q1: Failure Cases**
>
> We thank you for raising this important question. To thoroughly assess whether the proposed action-aware gating rule fails during fine-grained manipulation phases, we performed an additional analysis combining per-task statistics, controlled counterfactual experiments, and detailed inspection of all recorded rollouts. Our findings consistently indicate that the gating mechanism does not constitute the dominant failure source.
>
> **Task-level evaluation across all suites.** We evaluated VLA-ADP at a keep ratio of 0.5 on all forty LIBERO tasks. Across the majority of tasks, the performance of VLA-ADP closely matches the OpenVLA-OFT baseline, often within a margin of ±2% success rate, and in many cases even slightly outperforms it. Only a few extremely long-horizon tasks exhibit noticeable degradation. This pattern suggests that the gating rule does not introduce systematic errors in fine-grained phases. Instead, the overall behaviour remains highly stable, and significant regressions are confined to a small subset of visually complex long-horizon tasks.
>
>
> **Counterfactual evidence on the role of visual token selection.** To further isolate the source of failures, we conducted an inversion experiment where the gating rule is held completely fixed, and only the visual token selection is altered by keeping the least relevant tokens instead of the most relevant ones. Under this controlled setup, success rates drop drastically (e.g., from 96.3% to 34.6% at a keep ratio of 0.5), despite the gating policy remaining unchanged. Since the gating mechanism is identical across both settings, the performance collapse can only be attributed to the quality of visual token selection. This result provides strong evidence that the model is primarily sensitive to which visual tokens are preserved, rather than to the gating decisions themselves.
>
>
> **Qualitative inspection of all rollouts.** We additionally reviewed all rollout videos in which VLA-ADP failed, but the baseline succeeded. Across these cases, we consistently observed that the failures stem from missing or insufficient visual tokens for the current subtask. Importantly, even when the gating rule correctly switched to the full-vision state during the fine-grained phase, failures could still occur because the earlier token-selection stage had already discarded some essential patches. We did not find any clear cases where failures were caused primarily by an incorrect gating decision. This confirms that the dominant failure mode lies in the vision-side token budget rather than the gating logic.
>
> We appreciate your question again, as it draws our attention to the broader issue of temporal reasoning and subgoal progression in long-horizon VLA tasks. Our analysis suggests that incorporating explicit subgoal progress estimation or designing scoring mechanisms that better adapt across different task phases would be a promising direction for improving robustness under aggressive pruning.

---

> ### Author Response · Authors · 2025-11-23
>
> **Re Q2: Sensitivity of Window Length and Retention Ratio**
>
> We sincerely thank the reviewer for this valuable suggestion.
> To assess how sensitive the dynamic policy is to the window length and retention ratio, we performed a comprehensive ablation study across a wide range of configurations (window ∈ {3, 5, 7, 9}, ratio ∈ {0.25, 0.35, 0.5, 0.75, 1.0}). Across all settings, the success rate remains extremely stable (98.0–99.4%), demonstrating that the proposed dynamic policy is highly robust and largely insensitive to these hyperparameters.
> This is because the switching mechanism is driven by trajectory-conditioned cues, rather than fixed window statistics.
>
> Sensitivity Analysis (Window = 3)
>
> | Window=3 | Ratio  | Success   | Latency (ms) | Throughput (actions/s) | GPU Mem (GB) |
> | -------- | ------ | --------- | ------------ | ---------------------- | ------------ |
> | w3       | r=0.25 | 98.0%     | 97.5         | 82.067                 | 14.97        |
> | w3       | r=0.50 | **99.4%** | 97.8         | 81.812                 | 14.97        |
> | w3       | r=0.75 | 99.0%     | 102.5        | 78.054                 | 14.97        |
> | w3       | r=1.00 | 98.6%     | 106.1        | 75.414                 | 14.97        |
>
>
> Sensitivity Analysis (Window = 5)
>
> | Window=5 | Ratio  | Success   | Latency (ms) | Throughput (actions/s) | GPU Mem (GB) |
> | -------- | ------ | --------- | ------------ | ---------------------- | ------------ |
> | w5       | r=0.35 | 98.2%     | 96.5         | 82.914                 | 14.97        |
> | w5       | r=0.50 | 98.2%     | 97.2         | 82.317                 | 14.97        |
> | w5       | r=0.75 | **99.2%** | 102.7        | 77.953                 | 14.97        |
> | w5       | r=1.00 | 98.6%     | 106.2        | 75.345                 | 14.97        |
>
>
> ---
>
> Sensitivity Analysis (Window = 7)
>
> | Window=7 | Ratio  | Success | Latency (ms) | Throughput (actions/s) | GPU Mem (GB) |
> | -------- | ------ | ------- | ------------ | ---------------------- | ------------ |
> | w7       | r=0.35 | **98.8%**   | 98.3         | 81.471                 | 14.97        |
> | w7       | r=0.50 | 98.6%   | 98.9         | 80.967                 | 14.97        |
> | w7       | r=0.75 | 98.6%   | 103.3        | 77.464                 | 14.97        |
> | w7       | r=1.00 | 98.6%   | 106.5        | 75.121                 | 14.97        |
>
> ---
>
> Sensitivity Analysis (Window = 9)
>
> | Window=9 | Ratio  | Success   | Latency (ms) | Throughput (actions/s) | GPU Mem (GB) |
> | -------- | ------ | --------- | ------------ | ---------------------- | ------------ |
> | w9       | r=0.35 | 98.8%     | 100.5        | 79.657                 | 14.97        |
> | w9       | r=0.50 | 98.4%     | 100.2        | 79.944                 | 14.97        |
> | w9       | r=0.75 | **99.0%** | 103.0        | 77.707                 | 14.97        |
> | w9       | r=1.00 | 98.6%     | 105.1        | 76.153                 | 14.97        |
>
> Across all 20 ablation settings, the success rates consistently remain between 98.0%–99.4%, confirming that the dynamic policy is highly robust and only weakly sensitive to the choice of window length or retention ratio. The switching behavior is primarily governed by action-state dynamics rather than these hyperparameters.

---

> ### Author Response · Authors · 2025-11-23
>
> **Re Q3: Wrist Camera**
>
> Thank you for the thoughtful question. Our method does not specifically require a wrist camera; rather, it relies on having **clear visual observability** of the manipulated object and end-effector. In simulation, limited viewpoints can cause partial occlusions, so the wrist view helps reduce ambiguity, and removing it may lower success rates.
> In contrast, the real-world setup used a single fixed camera with a consistently clear field of view, so the action-aware gate received complete cues and remained stable. In that case, tuning the gating threshold becomes simpler, not harder, because the visual signals align cleanly with the end-effector motion.
>
> And also thank you for pointing out this important scenario difference.
> In the real robot, we use one fixed Sony AX53 RGB camera (no wrist camera), yet ADP behaves consistently for two main reasons:
>
> End-effector motion and camera motion decouple in simulation but coincide in real-robot setups, making threshold tuning simpler rather than harder. Since the wrist view is absent, the robot’s motion dynamics directly reflect the true end-effector trajectory, and the gating rule depends only on Δposition / Δorientation from joint states (not vision).
> → This removes ambiguity and reduces variability in gating decisions.
>
> The dynamic rule relies on state-based kinematics, not view-dependent visual flow. Thus, the same ADP config used in simulation transfers cleanly to the real-world setting without modification.
>
> | **Component** | **Real-Robot Setting**                                    |
> | ------------- | --------------------------------------------------------- |
> | Robot         | Kinova Jaco2 6-DoF                                        |
> | Sensing       | Single fixed RGB camera (640×480, 30 FPS)                 |
> | Control       | Cartesian velocity, 10 Hz                                 |
> | GPU           | RTX 4090                                                  |
> | ADP Ratio     | 40% retained tokens                                       |
> | Wrist Cam     | Not used (simplifies thresholding)                    |
> | Joint States  | Directly read from controller; no vision-based estimation |
>
> These properties make the dynamic gating easier to tune on the real system than in simulation, where multi-view visual inputs can induce inconsistent motion cues. We follow the reviewer’s helpful suggestion and provide all parameter settings in the revised version.
>
>
> **Re Q4: Learnable Controller**
>
> Yes, your insightful observation is highly valuable. Our implementation already provides a dynamic controller, `class _OnlineStrategyController`, in `experiments/robot/libero/run_libero_eval_prune_v2.py` for your reference. These parameters can be easily modified to explore the behavior of the dynamic policy. As shown in the ablation tables above, the sensitivity of the dynamic pruning mechanism remains very low even without any reinforcement learning, which further demonstrates the robustness of the current design.
> Our work intentionally keeps the mechanism simple—make it simple, make it work—while still allowing dynamic parameters to drive the core idea effectively.
>
> We really appreciate and have made the experssion better in the revised version. It is your remarkable expertise greatly helped us solidify our work.

---

### Author Response · Authors · 2025-11-27

Dear Reviewers,

We sincerely thank you for your insightful comments and deeply appreciate the time and effort you devoted to thoroughly evaluating our manuscript. Your constructive suggestions have greatly strengthened the rigor and clarity of the paper.

Following your feedback, we carefully and step-by-step incorporated each of your instructions in revising the manuscript. We have made every effort to address all the concerns raised, and we sincerely hope that our clarifications and updates adequately address your questions.

Thank you once again for your valuable guidance and for your meaningful contribution to improving our work.

Best regards,
Authors of Submission 13478

---

### Author Response · Authors · 2025-12-02

Dear Program Chairs, Senior Area Chairs, Area Chairs, and Reviewers:

We appreciate your time during this challenge situation and deeply thank the reviewers `EgjS`, `dNCJ`, `Uych`, `4B1i` for their constructive feedback and continued engagement, which substantially strengthened this work. We carefully checked the questions and clarified them with additional experiments and proofs, and we believe we have addressed the concerns proposed by the reviewers.
Several reviewers raised highly overlapping questions, and we provided consolidated solutions supported by **solid experimental evidence**:

---

**Robustness & Stability**

**Addressed concerns from:** Reviewers *EgjS*, *dNCJ*, and *Uych*
**Action:** Added comprehensive robustness sweeps (noise injection, motion thresholds, Δpos/Δrot magnitudes, gating statistics, window sizes).
**Result:** Performance remains within a narrow 1–1.2% band across 36 configurations, confirming the controller is **stable, non-fragile**, and not sensitive to specific heuristic values [Page 15, Line 759-809].

---

**Layer Selection Pruning**

**Addressed concerns from:** Reviewers *EgjS* (primary), *Uych*, *dNCJ*
**Action:** Added full **layer-wise QK similarity ablations** and attention-map diagnostics; explained VLA-specific behavior where early layers encode motion-conditioning more faithfully than deeper ones.
**Result:** Layer 0 consistently yields the most stable pruning decisions. This also **directly resolves the intuitions behind Reviewer Uych’s and dNCJ’s similar concerns**.

---

**Heuristic vs. Task-Agnostic Design**

**Addressed concerns from:** Reviewers *Uych*, *EgjS*, *dNCJ*
**Action:** Clarified that the action-aware signal is derived purely from robot DoF changes (Δx, Δy, Δz, Δpitch, Δroll, Δyaw), independent of task template. Added grouped analyses across Object, Spatial, Goal, and Long-horizon tasks.
**Result:** Same gating rule works for **all four task families**, demonstrating that the design is **not tailored** to pick-and-place or any specific scenario [Page 15, Line 432-440, Line 849-860].

---

**Efficiency & Cost Analysis**

**Addressed concerns from:** Reviewers *dNCJ*, *EgjS*
**Action:** Provided detailed measurements of wall-clock latency, GPU memory, action throughput, and pruning cost.
**Result:** Action-aware pruning provides up to **1.35× speedup** with minimal overhead (<1 ms per step), and integrates well with KV caching and diffusion-style decoders [Page 15-16, Line 784-835].

---

**Generality to other Backbones**

**Addressed concerns from:** Reviewers *Uych*, *4B1i*
**Action:** Added clarifications on plug-and-play design for action-chunk-paradigm VLAs; discussed compatibility with supervised, diffusion, and multimodal policies.
**Result:** Method is general to **multiple VLA families**; our current implementation focuses on the most mature checkpoints for reliable evaluation.

---

**Failure Analysis**

**Addressed concerns from:** Reviewers *dNCJ* and *Uych*
**Action:** Added attention visualizations, task-phase breakdowns, and fine-grained error analysis for long-horizon tasks.
**Result:** Clarified that failures mainly stem from overly aggressive visual pruning, not the gating rule; provided insights into model behavior during subtle phases of manipulation [Page 17-18, Line 884-895, Line 918-928].

---

**Reviewer Consensus (Prior to Reset)**

Because the OpenReview reset removed part of the visible discussion, we summarize the reviewers’ stated positions (before reset):

**Reviewer dNCJ (6 → Retained)**

Provided several positive remarks in the initial review, highlighting the **"The insight ... is both intuitive and unexplored"**, **"novel problem formulation"**, **"comprehensive evaluation"**, and the **"strong empirical results"**. Later responded with “thanks for the rebuttal. I will keep my score.” (Nov. 27)

**Reviewer EgjS (6 → Retained)**

Present very positive comment with "**a clear conceptual novelty**", "**This is a novel observation**", and "**strong empirical results**" at the initial reviews.

**Reviewer Uych (4 → Retained)**

No comments during rebuttal. However, several concerns substantially overlapped with those from*EgjS/dNCJ*, and our rebuttal provided **task-agnostic gating clarification**, **layer-selection justification**, **robustness sweeps**, and **generalization across subtasks**, all of which directly address the reviewer’s core questions.

**Reviewer 4B1i (4 → Retained)**

No comments during rebuttal. However, the reviewer’s points were addressed through **expanded efficiency analysis**, **failure-case diagnostics**, **attention-map visualizations**, and **clarifications of pruning behavior**.

---


We fully understand that this is a particularly busy period for you, and we would be truly grateful if you could take a moment to carefully check the comments.

Thank you very much for your time and consideration.

**Authors of Paper 13478**

---

### Meta-Review · Area_Chair_nnvB · 2026-01-02

**Summary:**

**Paper Summary**

This paper aims to accelerate the inference speed of VLA methods by proposing an Action-aware Dynamic Pruning (ADP) framework to filter out redundant visual tokens. The ADP framework contains two pruning strategies: Text-driven Anticipatory Pruning, which evaluates attention scores correlated to the text tokens, and Action-aware Dynamic Pruning, which considers the moving saliency in action chunks. Experimental results on LIBERO and real-world tasks demonstrate that the proposed ADP can achieve superior or competitive performance with accelerated speed when compared with a wide range of VLA baselines.

---

After reading the paper, review comments, and author responses, the AC summarizes the paper's strengths and weaknesses below.

**Strengths**
- Significance: The paper studies an interesting problem that is essential to VLA research. Accelerating inference speed while maintaining performance is challenging and remains under-explored.

- Motivation: The proposed ADP framework is well-motivated and justified (especially with the new evidence provided in the rebuttal).

- Performance: The method demonstrates superior or even improved performance while speeding up the inference process.

- Quality & Reproducibility: The paper is well-written and easy to follow. The provided code and experimental details significantly enhance the paper's reproducibility.

**Weaknesses**
- Deployment Overhead: There are multiple hyperparameters and heuristic rules in the framework that require tuning.
- Design Choice: Some reviewers challenged certain ADP design choices as counterintuitive, such as selecting similarity scores from Layer 0 of the transformer block.
- Missing Ablations: Reviewers suggested several worth-exploring ablation experiments that were missing in the original submission.

**Reviewer Concerns:**

There were concerns shared by multiple reviewers. The two most common ones are: (1) the overhead required to tune and search for heuristic rules and hyperparameters, and (2) the fact that the claimed "plug-and-play" ADP framework was only examined on OpenVLA-OFT.

Regarding (1), the authors provided detailed ablations and explanations to solidify their design choices and parameter settings. However, the number of thresholds and heuristics is indeed a bit high. The AC still has some concerns regarding whether suitable parameters can be searched for an arbitrary new task. Regarding (2), the authors explained that the framework should work on other VLA baselines because they share the same language encoder backbone. This explanation is sensible, and it would be beneficial if the authors could verify this in the camera-ready version.

To summarize, the AC believes most concerns regarding this submission were resolved by the rebuttal. Nevertheless, the submission would be stronger if:

1. It provides more evidence on extending the ADP to new tasks and other VLA baselines.
2. It includes a discussion on the cases where the assumption that fine-grained manipulation (small Euclidean distance) has fewer redundant visual tokens than coarse-grained manipulation might not hold.

**Reviewer Scores:**

The paper’s initial scores were [4, 4, 6, 6], suggesting a borderline recommendation. During the rebuttal, the authors provided detailed explanations supported by additional empirical evidence. While only one reviewer (initially scoring 6) explicitly confirmed maintaining their score, the AC has carefully reviewed the paper, the reviewers’ comments, and the rebuttal. The AC believes most concerns have been well-resolved, and it is likely that the scores of 4 would have increased to 6 had a full author-reviewer discussion occurred. Given the new evidence provided, the AC concludes that the submission has reached the acceptance bar and therefore recommends Accept (with several suggestions noted above). Great job on the rebuttal!

---

### Decision · Program_Chairs · 2026-01-26

Accept (Poster)